# Value Function in Frequency Domain and the Characteristic Value Iteration Algorithm

**Amir-massoud Farahmand**[*]
Vector Institute & University of Toronto
Toronto, Canada
farahmand@vectorinstitute.ai

## Abstract

This paper considers the problem of estimating the distribution of returns in reinforcement learning, i.e., distributional RL problem. It presents a new representational framework to maintain the uncertainty of returns and provides mathematical tools to compute it. We show that instead of representing a probability distribution function of returns, one can represent their characteristic function, the Fourier transform of their distribution. We call the new representation Characteristic Value Function (CVF). The CVF satisfies a Bellman-like equation, and its corresponding Bellman operator is contraction with respect to certain metrics. The contraction property allows us to devise an iterative procedure to compute the CVF, which we call Characteristic Value Iteration (CVI). We analyze CVI and its approximate variant and show how approximation errors affect the quality of the computed CVF.

## 1   Introduction

The object of focus of the conventional RL is the expected return of following a policy, i.e., the value function [Sutton and Barto, 2019]. The goal is to find a policy that maximizes that expectation over all states, i.e., the optimal policy. This leads to agents that do not consider the distribution of returns in their decision making, but only its first moment. This might be of concern in scenarios where the risk is of paramount importance. Estimating the distribution of the return facilitates designing agents that consider objectives more general than maximizing the expected return, such as various notions of risk [Tamar et al., 2012, Prashanth and Ghavamzadeh, 2013, García and Fernández, 2015, Chow et al., 2018].

The Distributional RL (DistRL) literature [Engel et al., 2005, Morimura et al., 2010b, Bellemare et al., 2017, Barth-Maron et al., 2018, Lyle et al., 2019], on the other hand, moves away from the conventional goal of estimating the expectation of return and attempts to estimate a richer representation of the return, such as the distribution itself [Morimura et al., 2010b,a] or some statistical functional of it [Rowland et al., 2018, Dabney et al., 2018, Rowland et al., 2019]. It is notable that so far the focus of the DistRL literature has mostly been on designing better performing agents according to the expected return, and not any risk-related performance measure, but it is conceivable that those methods can be be used for designing risk-aware agents too.

This paper develops a new framework for maintaining the information available in the distribution of returns. Instead of estimating the distribution function itself, we maintain the Characteristic Function (CF) of the returns. The CF of a random variable (r.v.) is the Fourier transform of its probability distribution function (PDF). Similar to PDF, the CF of a r.v. contains all the information available about the distribution of that r.v., i.e., CF and PDF have a bijection relationship. They are nonetheless

---

[*]Homepage: http://academic.sologen.net.

different representations of the uncertainty of a r.v., hence they allow different types of manipulations and processing. The benefit of a new representation is that it opens up the possibility of designing new algorithms. An example from the field of control theory is that we have both time and frequency domain representations of a dynamical system. Although they are equivalent in many cases, designing a controller in the frequency domain is sometimes easier and may provide better insights. This work brings the frequency-based representation of uncertainty to DistRL.

The estimation procedures based on CF are not novel. Methods based on the Empirical Characteristic Function (ECF) have a long history in the statistics and econometrics literature [Feuerverger and Mureika, 1977, Feuerverger and McDunnough, 1981, Feuerverger, 1990, Knight and Yu, 2002, Yu, 2004]. These methods are considered as alternatives to the maximum likelihood estimation (MLE), because as opposed to MLE, whose computation might be infeasible for some distributions, one can always define and compute the ECF. This paper is inspired from that literature and develops similar tools for RL and approximate dynamic programming.

The main idea of this work is that by transforming the return, which is a r.v., to the frequency domain through the Fourier transform, we can define Characteristic Value Function (CVF), which essentially captures all information about the distribution of the return. A contribution of this work is that we prove that CVF indeed satisfies a Bellman-like equation $\tilde{T}^\pi \tilde{V} = \tilde{V}$ (Section 3). The corresponding Bellman operator, however, is different from the conventional ones or those in the DistRL literature. Instead of having an additive form, it is multiplicative, i.e., $(\tilde{T}^\pi \tilde{V})(\omega; x) \triangleq \tilde{R}(\omega; x) \int \mathcal{P}^\pi(\mathrm{d}y|x)\tilde{V}(\gamma\omega; y)$ with $\omega$ being the frequency variable, $x$ being the state variable, and $\tilde{R}$ being the Fourier transform of the immediate reward distribution (we will define these quantities later). We also prove that the new Bellman operator is contraction with respect to (w.r.t.) some specific metrics defined in the frequency domain (Section 3.1). The contraction property suggests that one might find the CVF through an iterative procedure similar to value iteration, which we call the Characteristic Value Iteration (CVI) algorithm (Section 4). This is the algorithmic contribution of this work.

Any procedure that implements CVI, however, may not perform it exactly, for example because we only have data as opposed to the actual transition probability distribution or because the state space is very large and we need to use function approximation. In case we can only approximately perform CVI, which we call Approximate CVI (ACVI), we inevitably have some errors. To understand the effect of using function approximation on these errors better, we consider a class of band-limited (in the frequency domain) functions, and study their function approximation and covering number properties (in the extended version of the paper). Another contribution of this work is the analysis of how the errors caused at each iteration of ACVI propagate throughout iterations and affect the quality of the outcome CVF (Section 5). We show that the errors in earlier iterations decay exponentially fast, i.e., the past errors are forgotten quickly. This is the same phenomenon observed in the conventional approximate value iteration. Finally, we show how to convert the error of CVF in the frequency domain to an error in distributions, measured according to the $p$-smooth Wasserstein distance (Section 6).

## 2 Distributional Bellman equation

We consider a discounted Markov Decision Process (MDP) $(\mathcal{X}, \mathcal{A}, \mathcal{R}, \mathcal{P}, \gamma)$ [Szepesvári, 2010]. Here $\mathcal{X}$ is the state space, $\mathcal{A}$ is the action space, $\mathcal{P} : \mathcal{X} \times \mathcal{A} \to \mathcal{M}(\mathcal{X})$ is the transition probability kernel, $\mathcal{R} : \mathcal{X} \times \mathcal{A} \to \mathcal{M}(\mathbb{R})$ is the immediate reward distribution, and $0 \leq \gamma < 1$ is the discount factor.[2] The (Markov stationary) policy $\pi : \mathcal{X} \to \mathcal{M}(\mathcal{A})$ induces the transition probability kernel $\mathcal{P}^\pi : \mathcal{X} \to \mathcal{M}(\mathcal{X})$ and the immediate reward distribution for the policy $\mathcal{R}^\pi : \mathcal{X} \to \mathcal{M}(\mathbb{R})$.

An MDP together with an initial state distribution $\rho \in \mathcal{M}(\mathcal{X})$ encode the laws governing the temporal evolution of a discrete-time stochastic process controlled by an agent as follows: The controlled process starts at time $t = 0$ with random initial state $X_0$ drawn from $\rho$, i.e., $X_0 \sim \rho$. The agent following a policy $\pi$ chooses action $A_t \in \mathcal{A}$ according to $A_t \sim \pi(\cdot|X_t)$ (stochastic policy) or

$A_t = \pi(X_t)$ (deterministic policy). In response, the next state is $X_{t+1} \sim \mathcal{P}(\cdot|X_t, A_t)$ and the agent receives reward $R_t \sim \mathcal{R}(\cdot|X_t, A_t)$. This process repeats. We may occasionally use $R(x, a)$ or $R^\pi(x)$ to denote to the r.v. that is drawn from $\mathcal{R}(\cdot|x, a)$ or $\mathcal{R}^\pi(\cdot|x)$. Also we may use $z = (x, a)$ as a shorthand. When we refer to a r.v. $Z = (X, A)$, this should be interpreted as a r.v. defined with $A \sim \pi(\cdot|X)$, where the policy should be clear from the context.

The return of the agent starting from a state $x \in \mathcal{X}$ and following a policy $\pi$ is the following random variable:

$$G^\pi(x) = \sum_{i \geq 0} \gamma^i R_i.$$

The (conventional) value function $V^\pi$ is the first moment of this r.v., i.e.,

$$V^\pi(x) = \mathbb{E}\left[G^\pi(X_0)|X_0 = x\right].$$

Likewise, one may define the return $G^\pi(x, a)$ for starting from state $x$, choosing action $a$, and following policy $\pi$ afterwards. The corresponding first moment of $G^\pi(x, a)$ would be the action-value function $Q^\pi(x, a)$.

From $G^\pi(x) = R_0 + \gamma \sum_{i \geq 0} \gamma^i R_{i+1}$, we see that $G^\pi(x)$ is the addition of two r.v. $R_0$ and $\gamma G^\pi(X_1)$ with $X_1 \sim \mathcal{P}^\pi(\cdot|X_0 = x)$. Therefore, the law (probability distribution) of $G^\pi(x)$ is the same as the law of $R_0 + \gamma G^\pi(X_1)$, i.e.,

$$G^\pi(x) \overset{\text{(D)}}{=} R_0 + \gamma G^\pi(X_1). \tag{1}$$

Here we use the symbol $\overset{\text{(D)}}{=}$ to emphasize that we are comparing two probability distributions. This is the Bellman-like distributional equation in the conventional DistRL.

We can also have a similar equation that relates $\bar{G}^\pi$ (the distribution of the r.v. $G^\pi$) and $\bar{R}(x) = \mathcal{R}^\pi(\cdot|x)$ (the distribution of the r.v. $R^\pi(x)$) [Rowland et al., 2018]. To define it, we recall the definition of the pushforward measure: Given a probability distribution $\nu \in \mathcal{M}(\mathbb{R})$ and a measurable function $f : \mathbb{R} \to \mathbb{R}$, the pushforward measure $f_\# \nu \in \mathcal{M}(\mathbb{R})$ is defined as $(f_\# \nu)(A) = \nu(f^{-1}(A))$ for all Borel sets $A \subset \mathbb{R}$.

The Bellman operator $\bar{T}^\pi : \mathcal{M}(\mathcal{X}) \to \mathcal{M}(\mathcal{X})$ between distributions is defined as

$$(\bar{T}^\pi \bar{G})(x) \triangleq \int (r + \gamma y)_\# \bar{G}(y) \mathcal{R}^\pi(\mathrm{d}r|x) \mathcal{P}^\pi(\mathrm{d}y|x), \qquad \forall x \in \mathcal{X}.$$

With this notation, the distributional Bellman equation is

$$\bar{G}^\pi(x) = (\bar{T}^\pi \bar{G}^\pi)(x), \qquad \forall x \in \mathcal{X}. \tag{2}$$

The distributional Bellman equation represents the intrinsic uncertainty of the return due to the randomness of the dynamics and policy. We may occasionally use $\bar{V}^\pi$ to refer to $\bar{G}^\pi$, to show its close relation to the conventional value function.

## 3 Characteristic value function

The conventional approach to representing the uncertainty of a r.v. is through its probability distribution function. This is not the only way to characterize a r.v. though. An alternative is to characterize the r.v. through the Fourier transform of its distribution function. This is known as the Characteristic Function (CF) of the random variable [Williams, 1991].

In this section we show that the instead of representing the distribution function of the return $G^\pi$, we may represents its characteristic function. Interestingly, the CF of return satisfies a Bellman-like equation, which is quite different from the conventional ones (1) and (2) that we have encountered so far.

Let us briefly recall the definition of a CF of a random variable. Given a real-valued r.v. $X$ with the probability distribution $\mu \in \mathcal{M}(\mathbb{R})$, its corresponding CF $c_X : \mathbb{R} \to \mathbb{C}$ is the function defined as[3]

$$c_X(\omega) \triangleq \mathbb{E}\left[e^{jX\omega}\right] = \int \exp(jx\omega)\mu(\mathrm{d}x), \qquad \omega \in \mathbb{R} \tag{3}$$

where $j = \sqrt{-1}$ is the imaginary unit. The CF of a probability distribution is closely related to the Fourier transform of its distribution function. If the probability density function is well-defined, CF is its Fourier transform, though CF exists even if the density does not. Several properties of CF are summarized in an appendix of the extended version of the paper. Thinking in the terms of the spatial-frequency duality common in the Fourier analysis, the probability distribution function is the spatial representation of a r.v. (with the magnitude of the r.v. corresponding to the space dimension), and the CF is its frequency representation.

Consider the recursive relation $G^\pi(x) = R^\pi(x) + \gamma G^\pi(X')$, with $X' \sim \mathcal{P}^\pi(\cdot|x)$, between the return $G^\pi(x)$ (a r.v.) and the random reward $R^\pi(x)$ and the return at the next step $G^\pi(X')$. By the distributional equality of both sides (cf. (1)), we have

$$c_{G^\pi(x)}(\omega) = \mathbb{E}\left[\exp\left(j\omega G^\pi(x)\right)\right] = \mathbb{E}\left[\exp\left(j\omega\left(R^\pi(x) + \gamma G^\pi(X')\right)\right)\right], \qquad \forall \omega \in \mathbb{R}. \quad (4)$$

The right-hand side (RHS) of (4) is

$$\begin{aligned}
\mathbb{E}\left[\exp\left(j\omega\left(R^\pi(x) + \gamma G^\pi(X')\right)\right)\right] &= \mathbb{E}\left[\mathbb{E}\left[\exp\left(j\omega\left(R^\pi(x) + \gamma G^\pi(X')\right)\right) \mid X = x, A\right]\right] \\
&= \mathbb{E}\left[\mathbb{E}\left[\exp\left(j\omega R^\pi(x)\right) \mid X = x, A\right] \mathbb{E}\left[\exp\left(j\omega\gamma G^\pi(X')\right) \mid X = x, A\right]\right] \\
&= c_{R^\pi(x)}(\omega) \mathbb{E}\left[\mathbb{E}\left[\exp\left(j\omega\gamma G^\pi(X')\right) \mid X = x, A\right]\right] \\
&= c_{R^\pi(x)}(\omega) \mathbb{E}\left[\exp\left(j\omega\gamma G^\pi(X')\right) \mid X = x\right], \quad (5)
\end{aligned}$$

where $A$ is a r.v. drawn from $\pi(\cdot|x)$. Here we benefitted from the fact that the r.v. $R^\pi(x)$ and $G^\pi(X')$ are conditionally independent given $X = x$ and $A$.

Let us consider the CF of $G^\pi(X')$ conditioned on $X = x$:

$$\begin{aligned}
\mathbb{E}\left[\exp\left(j\omega G^\pi(X')\right) \mid X = x\right] &= \mathbb{E}\left[\mathbb{E}\left[\exp\left(j\omega G^\pi(X')\right) \mid X'\right] \mid X = x\right] \\
&= \int \mathcal{P}^\pi(\mathrm{d}x'|x)\mathbb{E}\left[\exp\left(j\omega G^\pi(x')\right)\right] \\
&= \mathbb{E}\left[c_{G^\pi(X')}(\omega) \mid X = x\right], \quad (6)
\end{aligned}$$

where we conditioned the inner expectation on the next-state $X'$ (so its randomness comes from the return from that point onward), and used the definition of CF.

Plugging (6) in (5) gives the RHS of (4). So we get

$$\begin{aligned}
c_{G^\pi(x)}(\omega) &= c_{R^\pi(x)}(\omega) \mathbb{E}\left[\exp\left(j\omega\gamma G^\pi(X')\right) \mid X = x\right] \\
&= c_{R^\pi(x)}(\omega)\mathbb{E}\left[c_{\gamma G^\pi(X')}(\omega) \mid X = x\right] \\
&= c_{R^\pi(x)}(\omega)\mathbb{E}\left[c_{G^\pi(X')}(\gamma\omega) \mid X = x\right] = c_{R^\pi(x)}(\omega) \int \mathcal{P}^\pi(\mathrm{d}y|x)c_{G^\pi(y)}(\gamma\omega), \quad (7)
\end{aligned}$$

where the penultimate equality is because of the scaling property of CF (refer to the extended version of the paper for more information).

We denote the CF of the reward $c_{R^\pi(x)}(\omega)$ by $\tilde{R}(\omega; x)$, and the CF of the return $c_{G^\pi(x)}(\omega)$ by $\tilde{V}^\pi(\omega; x)$ for all $x \in \mathcal{X}$ and $\omega \in \mathbb{R}$. Here the symbol $\tilde{\ }$ is used to remind us that we are referring to a CF of a random variable. With these notations, we can write (7) in more compact form of

$$\tilde{V}^\pi(\omega; x) = \tilde{R}(\omega; x) \int \mathcal{P}^\pi(\mathrm{d}y|x)\tilde{V}^\pi(\gamma\omega; y). \quad (8)$$

This is the Bellman-like equation between the CF of return and the reward. The function $\tilde{V}^\pi : \mathbb{R} \times \mathcal{X} \to \mathbb{C}_1$ (where $\mathbb{C}_1$ is the area within the unit circle in the complex plane, i.e., $\mathbb{C}_1 = \{z \in \mathbb{C} : |z| \leq 1\}$) is the CF of the $G^\pi(x)$ for all $x \in \mathcal{X}$. We call $\tilde{V}^\pi$ the Characteristic Value Function (CVF).

We also define the Bellman operator between the CF functions:

$$(\tilde{T}^\pi\tilde{V})(\omega; x) \triangleq \tilde{R}(\omega; x) \int \mathcal{P}^\pi(\mathrm{d}y|x)\tilde{V}(\gamma\omega; y).$$

With this notation, the Bellman equation can be written more compactly as

$$\tilde{V}^\pi = \tilde{T}^\pi\tilde{V}^\pi.$$

It is worth mentioning that for any fixed $x \in \mathcal{X}$, $\omega \mapsto \tilde{V}^\pi(\omega; x)$ is a CF. A CF is continuous function of $\omega$ and its magnitude is bounded by 1 (refer to the extended version of the paper).

## 3.1 Bellman operator is contraction

We show that the Bellman operator $\tilde{T}^\pi$ is a contraction w.r.t. certain metrics, to be specified. This allows us to devise a value iteration-like procedure that converges to the CVF $\tilde{V}^\pi$ of a policy $\pi$.

We first define some distance metrics between CFs. Given two CF $c_1, c_2 : \mathbb{R} \to \mathbb{C}$, and $p \geq 1$, we define

$$d_{\infty,p}(c_1, c_2) \triangleq \sup_{\omega \in \mathbb{R}} \left| \frac{c_1(\omega) - c_2(\omega)}{\omega^p} \right|, \qquad d_{1,p}(c_1, c_2) \triangleq \int \left| \frac{c_1(\omega) - c_2(\omega)}{\omega^p} \right| d\omega. \qquad (9)$$

Here we use the convention that $\frac{0}{0} = 0$.[4]

We also define similar metrics for functions such as $\tilde{R}$ and $\tilde{V}^\pi$. Given $\tilde{V}_1, \tilde{V}_2 : \mathbb{R} \times \mathcal{X} \to \mathbb{R}$, we define

$$d_{\infty,p}(\tilde{V}_1, \tilde{V}_2) \triangleq \sup_{x \in \mathcal{X}} \sup_{\omega \in \mathbb{R}} \left| \frac{\tilde{V}_1(\omega; x) - \tilde{V}_2(\omega; x)}{\omega^p} \right|, \quad d_{1,p}(\tilde{V}_1, \tilde{V}_2) \triangleq \sup_{x \in \mathcal{X}} \int \left| \frac{\tilde{V}_1(\omega; x) - \tilde{V}_2(\omega; x)}{\omega^p} \right| d\omega. \qquad (10)$$

There are similar to the distances for comparing two CFs, with the difference that we take the supremum over all states $x \in \mathcal{X}$. To be more precise about how the distances are calculated (e.g., sup over $\mathcal{X}$, etc.), we could use $d_{\mathcal{X}(\infty),\omega(\infty,p)}(\tilde{V}_1, \tilde{V}_2)$ instead of $d_{\infty,p}(\tilde{V}_1, \tilde{V}_2)$. To simplify the notations, however, we use the overloaded symbols $d_{\infty,p}$ and $d_{1,p}$ instead.

Based on these distances, we define the following norms for a function $\tilde{V} : \mathbb{R} \times \mathcal{X} \to \mathbb{R}$

$$\left\| \tilde{V} \right\|_{\infty,p} = d_{\infty,p}(\tilde{V}, 0), \qquad \left\| \tilde{V} \right\|_{1,p} = d_{1,p}(\tilde{V}, 0),$$

where $0$ is a constant function $(\omega; x) \mapsto 0$. We sometimes refer to the supremum w.r.t. $x \in \mathcal{X}$ of $\tilde{V}$ by $\|\tilde{V}(\omega; \cdot)\|_\infty = \sup_{x \in \mathcal{X}} |\tilde{V}(\omega; x)|$. This should not be confused with $\|\tilde{V}\|_{\infty,p}$, whose supremum is over both $\omega$ and $x$, and the $\omega$ variable is weighted by $w^{-p}$.

Several properties of $d_{\infty,p}$ and $d_{1,p}$ are presented in an appendix of the extended version of the paper. Briefly, we show that $d_{1,p}$ and $d_{\infty,p}$ are metrics. We also show that the space of VCFs $\mathcal{V} = \{\tilde{V} : \mathbb{R} \times \mathcal{X} \to \mathbb{C}_1 : \tilde{V}(0; x) = 1\}$, which is a superset of the space of all feasible VCFs, endowed with $d_{\infty,p}$ is complete.

The following result shows that the Bellman operator for VCF is a contraction operator w.r.t. $d_{1,p}$ and $d_{\infty,p}$. This is the main result of this section.

**Lemma 1.** *Let $0 < \gamma < 1$. The operator $\tilde{T}^\pi$ is a $\gamma^p$-contraction in $d_{\infty,p}$ (for $p > 0$) and $\gamma^{p-1}$-contraction in $d_{1,p}$ (for $p > 1$). That is, for any $\tilde{V}_1, \tilde{V}_2 : \mathbb{R} \times \mathcal{X} \to \mathbb{C}$ with $d_{\infty,p}(\tilde{V}_1, \tilde{V}_2) < \infty$ or $d_{1,p}(\tilde{V}_1, \tilde{V}_2) < \infty$, we have*

$$d_{\infty,p}(\tilde{T}^\pi \tilde{V}_1, \tilde{T}^\pi \tilde{V}_2) \leq \gamma^p d_{\infty,p}(\tilde{V}_1, \tilde{V}_2),$$
$$d_{1,p}(\tilde{T}^\pi \tilde{V}_1, \tilde{T}^\pi \tilde{V}_2) \leq \gamma^{p-1} d_{1,p}(\tilde{V}_1, \tilde{V}_2).$$

For the contraction to be non-trivial, and avoid having a trivial inequality such as $\infty \leq \gamma^p \infty$, we require the boundedness of $d_{\infty,p}(\tilde{V}_1, \tilde{V}_2)$ or $d_{1,p}(\tilde{V}_1, \tilde{V}_2)$. This is a condition that should be verified, and as we shall soon see holds under certain conditions.

We briefly remark that the Bellman operator $\tilde{T}^\pi$ is not a contraction w.r.t. the supremum norm $\|\tilde{V}\|_\infty = \sup_{x \in \mathcal{X}} \sup_{\omega \in \mathbb{R}} |\tilde{V}(\omega; x)|$. This is shown in the extended version of the paper.

The importance of showing that the Bellman operator for VCF is a contraction is that we can then apply the Banach fixed point theorem (e.g., Theorem 3.2 of Hunter and Nachtergaele [2001]) to show the uniqueness of the fixed point $\tilde{V}^\pi$ (we also require the completeness of the space, which is shown for $d_{\infty,p}$). Moreover, it suggests that we can find the fixed point by iterative application of the operator. This is the path we pursue in the next section.

# 4 Characteristic value iteration

The contraction property of the Bellman operator $\tilde{T}^\pi$ (Lemma 1) suggests that we can find $\tilde{V}^\pi$ by an iterative procedure, similar to the conventional value iteration. The procedure is

$$\tilde{V}_1 \leftarrow \tilde{R},$$
$$\tilde{V}_{k+1} \leftarrow \tilde{T}^\pi \tilde{V}_k = \tilde{R}\mathcal{P}^\pi \tilde{V}_k. \qquad (k \geq 1) \qquad (11)$$

We call this procedure Characteristic Value Iteration (CVI).

CVI converges under certain conditions. To see this, notice that $\tilde{V}^\pi = \tilde{T}^\pi \tilde{V}^\pi$, so for $p \geq 1$ by Lemma 1 we have

$$d_{\infty,p}(\tilde{T}^\pi \tilde{V}_k, \tilde{V}^\pi) = d_{\infty,p}(\tilde{T}^\pi \tilde{V}_k, \tilde{T}^\pi \tilde{V}^\pi) \leq \gamma^p d_{\infty,p}(\tilde{V}_k, \tilde{V}^\pi),$$

under the condition that $d_{\infty,p}(\tilde{V}_k, \tilde{V}^\pi) < \infty$. Similarly, we have $d_{1,p}(\tilde{T}^\pi \tilde{V}_k, \tilde{V}^\pi) \leq \gamma^{p-1} d_{1,p}(\tilde{V}_k, \tilde{V}^\pi)$ (for $p > 1$). By the iterative application of this upper bound, assuming that $d_{\infty,p}(\tilde{R}, \tilde{V}^\pi) < \infty$, we get that

$$d_{\infty,p}(\tilde{V}_{k+1}, \tilde{V}^\pi) \leq \gamma^p d_{\infty,p}(\tilde{V}_k, \tilde{V}^\pi) \leq \cdots \leq (\gamma^p)^k d_{\infty,p}(\tilde{V}_1, \tilde{V}^\pi) = (\gamma^p)^k d_{\infty,p}(\tilde{R}, \tilde{V}^\pi). \quad (12)$$

Likewise, assuming that $d_{1,p}(\tilde{R}, \tilde{V}^\pi) < \infty$, we obtain

$$d_{1,p}(\tilde{V}_{k+1}, \tilde{V}^\pi) \leq (\gamma^{p-1})^k d_{1,p}(\tilde{R}, \tilde{V}^\pi). \qquad (13)$$

As long as $d_{\infty,p}(\tilde{R}, \tilde{V}^\pi)$ (or $d_{1,p}(\tilde{R}, \tilde{V}^\pi)$) is finite for some $p \geq 1$ ($p > 1$), CVI converges geometrically fast. A result in an appendix of the extended version of the paper specifies the condition when the $d_{\infty,p}$ distance of two CF would be finite. For $p = 1$, it is sufficient that the immediate reward $R^\pi(x) \sim \mathcal{R}(\cdot; x)$ and the return $G^\pi(\cdot; x)$ be integrable, i.e., $\mathbb{E}\left[|R^\pi(x)|\right], \mathbb{E}\left[|G^\pi(\cdot; x)|\right] < \infty$ for all states $x \in \mathcal{X}$. Since we deal with discounted MDP, the integrability of $R^\pi(x)$ (uniformly over $\mathcal{X}$) entails the integrability of $G^\pi(\cdot; x)$. Therefore under very mild conditions, CVI is convergent w.r.t. $d_{\infty,1}$.

For integer valued $p \geq 2$, the condition becomes more restrictive. The first requirement is that $\mathbb{E}\left[|R^\pi(x)|^p\right]$ and $\mathbb{E}\left[|G^\pi(\cdot; x)|^p\right]$ are finite. This is not restrictive, and holds for many problems. The restrictive condition is that the first $k = 1, \ldots, p-1$ moments of the reward and the return should match, i.e., $\mathbb{E}\left[R^\pi(x)^k\right] = \mathbb{E}\left[G^\pi(x)^k\right]$ for all $x \in \mathcal{X}$. This does not seem realistic, perhaps except for $p = 2$ when problems with zero expected immediate reward for all states but with varying variance are imaginable.

One can show that the fixed point of $\tilde{T}^\pi$ is unique. The result is formally stated in the extended version of the paper.

## 4.1 Approximate characteristic value iteration

Performing CVI (11) exactly may not be practical, for at least two reasons. First, for problems with large state space, we cannot represent $\tilde{V}^\pi$ exactly and we need to rely on function approximation. Second, for learning scenario where we do not have access to the model $\mathcal{P}^\pi$, but only observe data from interacting with the environment, we cannot apply the Bellman operator $\tilde{T}^\pi$ exactly either.

We can extend CVI to Approximate CVI (ACVI) similar to how exact VI can be extended to Approximate Value Iteration, also known as Fitted Value Iteration or Fitted Q-Iteration. Various variants of AVI have been empirically and theoretically studied in the literature [Ernst et al., 2005, Munos and Szepesvári, 2008, Farahmand et al., 2009, Silver et al., 2016, Tosatto et al., 2017, Chen and Jiang, 2019]. We would like to build the same general framework for CVF and CVI.

Suppose that for whatever reason we perform each iteration of CVI only approximately, that is, $\tilde{V}_{k+1} \approx \tilde{T}^\pi \tilde{V}_k$. The resulting procedure can be described as

$$\tilde{V}_1 \leftarrow \tilde{R} + \tilde{\varepsilon}_1,$$
$$\tilde{V}_{k+1} \leftarrow \tilde{T}^\pi \tilde{V}_k + \tilde{\varepsilon}_{k+1}. \qquad (k \geq 1) \qquad (14)$$

Here $\tilde{\varepsilon}_k : \mathbb{R} \times \mathcal{X} \to \mathbb{C}$ is the error in the frequency-state space. Recall that the value of a valid CF at frequency $\omega = 0$ is equal to one, i.e., $c(0) = 1$. To ensure that $\tilde{V}_k(\cdot; x)$ is a CF for all $x \in \mathcal{X}$, we must have $\tilde{V}_k(0; x) = 1$. This is satisfied if we require that $\tilde{\varepsilon}_k(0; x) = 0$ for all $k = 1, 2, \ldots$ and $x \in \mathcal{X}$. We can interpret this requirement by noticing that the condition $c(0) = 1$ is simply a requirement that $c(0) = \mathbb{E}\left[e^{jX0}\right] = \mathbb{E}\left[1\right] = \int \mu(\mathrm{d}x)$ be equal to 1. So we are essentially requiring that we do not lose or add probability mass at each iteration of ACVI.

Performing ACVI can be quite similar to the conventional AVI. Suppose that we are given a dataset $\mathcal{D}_n = \{(X_i, R_i, X_i')\}_{i=1}^n$, with $X_i \sim \mu$, $X_i' \sim \mathcal{P}^\pi(\cdot|X_i)$ and $R_i \sim \mathcal{R}^\pi(\cdot|X_i)$. Given this dataset and a CVF $\tilde{V}$, we define the empirical Bellman operator as the following mapping:

$$(\hat{\tilde{T}}^\pi \tilde{V})(\omega; X_i) \triangleq e^{j\omega R_i} \tilde{V}(\gamma\omega; X_i'), \qquad \forall \omega \in \mathbb{R}, \forall i = 1, \ldots, n.$$

For any fixed function $\tilde{V}$ and at any fixed state $X_i$, with a r.v. $A_i \sim \pi(\cdot|X_i)$, we have

$$\mathbb{E}\left[(\hat{\tilde{T}}^\pi \tilde{V})(\omega; X) \mid X = X_i\right] = \mathbb{E}\left[e^{j\omega R_i} \tilde{V}(\gamma\omega; X_i') \mid X = X_i\right]$$
$$= \tilde{R}(\omega; X_i) \int \mathcal{P}^\pi(\mathrm{d}y|X_i)\tilde{V}(\gamma\omega; y) = (\tilde{T}^\pi \tilde{V})(\omega; X_i).$$

This shows that the random process $(\hat{\tilde{T}}^\pi \tilde{V})(\omega; X_i)$ is an unbiased estimate of $(\tilde{T}^\pi \tilde{V})(\omega; X_i)$. In other words, $(\tilde{T}^\pi \tilde{V})(\omega; X_i)$ is the conditional mean of $(\hat{\tilde{T}}^\pi \tilde{V})(\omega; X_i)$. Finding the conditional mean of a r.v. is the regression problem (i.e., estimating $m(x) = \mathbb{E}[Y|X = x]$ by $\hat{m}(x)$ using a dataset of $\{(X_i, Y_i)\}_{i=1}^n$), which has been extensively studied in the statistics and machine learning literature [Györfi et al., 2002, Wasserman, 2007, Hastie et al., 2009, Goodfellow et al., 2016]. A powerful estimator that generalizes well across states and $\omega$ allows us to approximately perform one step of ACVI.

One approach to finding a regression estimator is to solve an empirical risk minimization problem:

$$\tilde{V}_{k+1} \leftarrow \underset{\tilde{V} \in \mathcal{F}}{\mathrm{argmin}} \frac{1}{n} \sum_{i=1}^n \int \left|\tilde{V}(\omega; X_i) - e^{j\omega R_i} \tilde{V}_k(\gamma\omega; X_i')\right|^2 w(\omega)\mathrm{d}\omega, \tag{15}$$

where $\mathcal{F} \subset \mathcal{V}$ is a space of functions from $\mathbb{R} \times \mathcal{X}$ to $\mathbb{C}_1$, which can be represented by various types of function approximators (including decision trees, kernel-based ones, and neural networks), and $w : \mathbb{R} \mapsto \mathbb{R}$ is a weighting function that indicates the importance of different frequencies $\omega$. This is similar to the usual Fitted Value Iteration procedure [Ernst et al., 2005, Munos and Szepesvári, 2008, Farahmand et al., 2009, Silver et al., 2016, Tosatto et al., 2017, Chen and Jiang, 2019], which solves

$$V_{k+1} \leftarrow \underset{V \in \mathcal{F}}{\mathrm{argmin}} \frac{1}{n} \sum_{i=1}^n \left|V(X_i) - (R_i + \gamma V_k(X_i'))\right|^2, \tag{16}$$

with appropriately chosen function space $\mathcal{F}$ (and similar for Fitted Q Iteration and the action-value function $Q$). One clear difference between (15) and (16) is that we have an integral over the frequency domain in the former. This one-dimensional integral can be numerically integrated, for example, by discretizing the low-frequency domain $[-b, +b]$ (with $b > 0$) with resolution $\varepsilon_{\mathrm{int}}$. This incurs some controlled numerical error that is a function of $\varepsilon_{\mathrm{int}}$. For some function approximators, such as a decision tree, one might be able to calculate the integral more efficiently by benefitting from the constancy of values within a leaf.

The quality of approximating $\tilde{T}^\pi \tilde{V}_k$ by $\tilde{V}_{k+1}$ determines the error $\tilde{\varepsilon}_k$. The error depends on the regression method being used, as well as the number of data points available, capacity and expressibility of the function space $\mathcal{F}$, etc. We do not analyze this regression problem in this paper. We are nevertheless interested in knowing whether one can hope to have a small error with a reasonably selected $\mathcal{F}$. Two relevant questions are whether one can approximate $\tilde{T}^\pi \tilde{V}_k$ within $\mathcal{F}$ well enough (function approximation error), and whether $\mathcal{F}$ has enough regularity to allow reasonable convergence rate for the estimation error. We study these questions in detail in the appendices of the extended version of the paper. We only briefly mention that if the reward distribution is smooth in a certain sense, a band-limited function class $\mathcal{F}_b = \{\tilde{V} : \mathbb{R} \times \mathcal{X} \to \mathbb{C}_1 : \tilde{V}(0; x) = 1, \tilde{V}(\omega; x) = 0 \ \forall |\omega| > b\}$

provides an approximation error that goes to zero as the bandwidth $b$ increases. More specifically, the $d_{\infty,1}$ distance-based norm of the approximation error behaves like $O(b^{-\frac{1}{1+\beta}})$ with $\beta$ being the smoothness parameter. Furthermore, if the first $s$ absolute moments of the reward distribution are finite, the CVF $\tilde{V}(\cdot;x)$ belongs to the smoothness class $C^s([-b,b]) \cap \mathcal{F}_b$. This leads to a well-behaving covering number, which can be used to obtain a convergence rate for the estimation error. A side benefit of working with a band-limited function space is that the integral in (15) can be converted to a definite integral, which is easier to integrate numerically.

Next we analyze how these errors, however generated, affect the quality of the outcome $\tilde{V}_K$ after performing $K$ steps of ACVI.

## 5 Error propagation analysis

We analyze how the errors in the ACVI procedure (14) propagate throughout the iterations and affect the quality of the outcome CVF $\tilde{V}_K$, where $K$ is the number of times the iteration is performed.

We skip all the intermediate steps required to prove the main result of this section. They can be found in the same section of the extended version of the paper.

**Theorem 2.** *Consider the ACVI procedure* (14) *after $K \geq 1$ iterations. Assume that $\tilde{\varepsilon}_k(0;x) = 0$ for all $x \in \mathcal{X}$ and $k = 1, \ldots, K+1$. We have*

$$d_{\infty,p}(\tilde{V}_{K+1}, \tilde{V}^\pi) \leq \sum_{i=0}^{K} (\gamma^p)^i \|\tilde{\varepsilon}_{K+1-i}\|_{\infty,p} + (\gamma^p)^K d_{\infty,p}(\tilde{R}, \tilde{V}^\pi), \qquad (p \geq 1)$$

$$d_{1,p}(\tilde{V}_{K+1}, \tilde{V}^\pi) \leq \sum_{i=0}^{K} (\gamma^{p-1})^i \|\tilde{\varepsilon}_{K+1-i}\|_{1,p} + (\gamma^{p-1})^K d_{1,p}(\tilde{R}, \tilde{V}^\pi). \qquad (p > 1)$$

This result shows how the errors $\tilde{\varepsilon}_k$ in the ACVI procedure propagate throughout iterations and affect the quality of the approximation of $\tilde{V}^\pi$ by $\tilde{V}_{K+1}$. The error is measured according to the distances $d_{1,p}$ and $d_{\infty,p}$. The upper bounds show that errors in the earlier iterations are geometrically decayed. This entails that if the resources are limited, it is better to ensure the smallness of errors in later iterations. This phenomenon is similar to what we have observed in the conventional value iteration [Farahmand et al., 2010].

As discussed in Section 4, the condition that $d_{\infty,p}(\tilde{R}, \tilde{V}^\pi)$ is finite might be very restrictive for $p > 2$ and even for $p = 2$, it might hold only in special problems. But the finiteness of $d_{\infty,1}$ requires mild conditions. For the finiteness of $d_{\infty,1}(\tilde{R}, \tilde{V}^\pi)$ in the upper bound, the finiteness of the first absolute moment of the reward function is sufficient, as discussed after (13). For the finiteness of $\|\tilde{\varepsilon}_i\|_{\infty,1}$ terms, it is sufficient that $\tilde{\varepsilon}_i(0;x) = 0$ and that its first derivative w.r.t. $\omega$ is bounded for all states $x \in \mathcal{X}$, i.e., $|\tilde{\varepsilon}^{(1)}(\omega;x)| < \infty$. Based on these, so from now on we focus on $p = 1$.

## 6 From error in frequency domain to error in probability distributions

Theorem 2 in the previous section relates the errors at each iteration of ACVI to the quality of the obtained approximation of $\tilde{V}^\pi$. The error is measured according to the metrics $d_{1,p}$ and $d_{\infty,p}$. These are metrics in the frequency domain. What does having a small error in the frequency domain imply about the quality of approximating the distribution of returns $\bar{V}^\pi$?

From Levy's continuity theorem we know that the pointwise convergence of CF implies the convergence in distribution of their corresponding distributions. This suggest that we could define the error in the frequency domain

$$d_{\text{unif}}(\tilde{V}, \tilde{V}^\pi) = \sup_{x \in \mathcal{X}} \sup_{\omega \in \mathbb{R}} \left| \tilde{V}(\omega;x) - \tilde{V}^\pi(\omega;x) \right|.$$

Nevertheless, we did not define the distance this way because the Bellman operator would not be a contraction w.r.t. to it. So a valid question is whether, or in what sense, the smallness of $d_{\infty,p}(\tilde{V}, \tilde{V}^\pi)$ implies anything about the closeness of their corresponding probability distribution functions $\bar{V}$

and $\bar{V}^{\pi}$? In this section we show that such a relation indeed exists. We relate $d_{\infty,p}$ and $d_{1,p}$ to the $p$-smooth Wasserstein distance of the probability distribution functions [Arras et al., 2017].

**Definition 1.** *Let $p \geq 1$, $\mathcal{C}^p(\Omega)$ be the space of $p$-times continuous differentiable functions on domain $\Omega$, and $\mathcal{F}_p(\Omega) = \left\{ f \in \mathcal{C}^p(\Omega) : \|f^{(k)}\|_{\infty} \leq 1, 0 \leq k \leq p \right\}$. For two probability distributions $\mu_1, \mu_2 \in \mathcal{M}(\Omega)$, the $p$-smooth Wasserstein distance is defined as*

$$\mathcal{W}_{\mathcal{C}_p}(\mu_1, \mu_2) = \sup_{f \in \mathcal{F}_p(\Omega)} \left| \int f(x) \left( \mathrm{d}\mu_1(x) - \mathrm{d}\mu_2(x) \right) \right|.$$

*Remark* 1. Note that the conventional 1-Wasserstein distance is defined as

$$\mathcal{W}_1(\mu_1, \mu_2) = \sup_{f \in \mathrm{Lip}_1(\Omega)} \left| \int f(x) \left( \mathrm{d}\mu_1(x) - \mathrm{d}\mu_2(x) \right) \right|,$$

where $\mathrm{Lip}_1$ is the space of 1-Lipschitz functions. As $\|f^{(1)}\|_{\infty} \leq 1$ implies 1-Lipschitz functions, but not necessarily vice versa, $\mathcal{W}_{\mathcal{C}_1}(\mu_1, \mu_2) \leq \mathcal{W}_1(\mu_1, \mu_2)$.

Let us also define the $p$-smooth Wasserstein between $\bar{V}_1$ and $\bar{V}_2$ as follows:

$$\mathcal{W}_{\mathcal{C}_p}(\bar{V}_1, \bar{V}_2^{\pi}) \triangleq \sup_{x \in \mathcal{X}} \mathcal{W}_{\mathcal{C}_p}(\bar{V}_1(\cdot; x), \bar{V}_2^{\pi}(\cdot; x)).$$

This is the maximum over states $x \in \mathcal{X}$ of the value of the $p$-smooth Wasserstein between the distribution of return according to the probability distributions $\bar{V}_1(\cdot; x)$ and $\bar{V}_2(\cdot; x)$.

**Theorem 3.** *Consider the ACVI procedure (14) after $K \geq 1$ iterations. Assume that $\tilde{\varepsilon}_k(0; x) = 0$ for all $x \in \mathcal{X}$ and $k = 1, \dots, K+1$. Furthermore, assume that the immediate reward distribution $\mathcal{R}^{\pi}(\cdot | x)$ is $R_{max}$-bounded. We then have*

$$\mathcal{W}_{\mathcal{C}_2}(\bar{V}_{K+1}, \bar{V}^{\pi}) \leq \frac{2\sqrt{2}}{\sqrt{\pi}} \sqrt{\frac{R_{max}}{1 - \gamma}} \left[ \sum_{i=0}^{K} \gamma^i \|\tilde{\varepsilon}_{K+1-i}\|_{\infty,1} + \frac{2\gamma^K}{1 - \gamma} R_{max} \right].$$

This upper bound can be simplified if we are willing to provide a uniform over iterations upper bound on $\|\tilde{\varepsilon}_{K+1-i}\|_{\infty,1}$. In that case, we have

$$\mathcal{W}_{\mathcal{C}_2}(\bar{V}_{K+1}, \bar{V}^{\pi}) \leq \frac{2\sqrt{2R_{\max}}}{\sqrt{\pi}(1 - \gamma)^{3/2}} \left[ \max_{i=1,\dots,K+1} \|\tilde{\varepsilon}_i\|_{\infty,1} + 2\gamma^K R_{\max} \right].$$

We note that the 2-smooth Wasserstein distance $\mathcal{W}_{\mathcal{C}_2}$, which is an integral probability metric [Müller, 1997], is only one of the many distances between probability distributions [Gibbs and Su, 2002]. The choice of the right probability distance most likely depends on the performance measure we would like the policy to optimize. Studying this further is an interesting topic of future research.

## 7 Conclusion

This paper laid the groundwork for a new class of distributional RL algorithms. We have shown that one might represent the uncertainty about the return in the frequency domain, and such a representation (called Characteristic Value Function) enjoys properties such as satisfying a Bellman equation and having a contractive Bellman operator. This in turn allows us to compute the CVF by an iterative method called Characteristic Value Iteration. We also showed the effect of errors in the iterative procedure, and provided error propagation results, in both the frequency domain and the probability distribution space.

This paper is only the first step towards understanding CVFs and their properties. Among remaining questions is how to perform the regression step (15) of ACVI properly and efficiently. Specifically, how should we set the weighting function $w(\omega)$ in order to achieve accurate CVF in frequencies that are relevant for the tasks we want to solve. Studying other distances between CFs and their properties is another interesting research directions. This work only focused on the policy evaluation problem, so another obvious direction is designing risk-aware policy optimization algorithms based on CVF. Finally, empirically evaluating this approach for return uncertainty representation may lead to better understanding of its strengths and weaknesses.

**Acknowledgments**

I would like to thank the anonymous reviewers for their helpful feedback, particularly Reviewer #4. I acknowledge the funding from the Canada CIFAR AI Chairs program.

## Footnotes

[2]Here $\mathcal{M}(\Omega)$ refers to the space of all probability distributions on an appropriately defined $\sigma$-algebra of $\Omega$, e.g., the Borel $\sigma$-algebra on $\mathbb{R}$. We do not deal with the measure theoretic considerations in this work. Refer to Appendix C of Bertsekas [2013] or Chapter 7 of Bertsekas and Shreve [1978]. We occasionally use $\bar{X}$ to denote the probability distribution $\mu$ of the r.v. $X$.

[3]Here $X$ is a generic r.v. and does not refer to the state. The particular r.v. will be clear from the context.

[4]The metric $d_{\infty,p}$ has been studied under the name of Fourier-based metric Carrillo and Toscani [2007], and is called Toscani distance by Villani [2008].

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
