[Supplementary Material · CVI(NeurIPS2019)(extended).pdf]

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

We first prove the result for $d_{\infty,p}$. For any $x \in \mathcal{X}$, we have

$$d_{\infty,p}\left((\tilde{T}^\pi \tilde{V}_1)(\cdot; x), (\tilde{T}^\pi \tilde{V}_2)(\cdot; x)\right) = \sup_\omega \left|\frac{(\tilde{T}^\pi \tilde{V}_1)(\omega; x) - (\tilde{T}^\pi \tilde{V}_2)(\omega; x)}{\omega^p}\right|$$

$$\leq \sup_\omega \left[\underbrace{|\tilde{R}(\omega; x)|}_{\leq 1} \int \mathcal{P}^\pi(\mathrm{d}y|x) \frac{\left|\tilde{V}_1(\gamma\omega; y) - \tilde{V}_2(\gamma\omega; y)\right|}{|\omega|^p}\right]$$

$$\leq \int \mathcal{P}^\pi(\mathrm{d}y|x) \sup_\omega \left|\frac{\tilde{V}_1(\gamma\omega; y) - \tilde{V}_2(\gamma\omega; y)}{|\omega|^p}\right|. \tag{11}$$

We benefited from $|\tilde{R}(\omega; x)| \leq 1$ to get rid of the term depending of $\tilde{R}$, see Lemma 8 in Appendix A.
Denote $\nu = \gamma\omega$, and write

$$\sup_{\omega \in \mathbb{R}} \left|\frac{\tilde{V}_1(\gamma\omega; y) - \tilde{V}_2(\gamma\omega; y)}{|\omega|^p}\right| = \sup_{\nu \in \mathbb{R}} \left|\frac{\tilde{V}_1(\nu; y) - \tilde{V}_2(\nu; y)}{|(\frac{\nu}{\gamma})|^p}\right| = |\gamma|^p \sup_{\nu \in \mathbb{R}} \left|\frac{\tilde{V}_1(\nu; y) - \tilde{V}_2(\nu; y)}{|\nu|^p}\right|$$

$$= |\gamma|^p d_{\infty,p}\left(\tilde{V}_1(\cdot; y), \tilde{V}_2(\cdot; y)\right). \tag{12}$$

Plugging this result in the RHS of (11), and taking the supremum over $y$, we get that

$$d_{\infty,p}\left((\tilde{T}^\pi \tilde{V}_1)(\cdot; x), (\tilde{T}^\pi \tilde{V}_2)(\cdot; x)\right) \leq \gamma^p \int \mathcal{P}^\pi(\mathrm{d}y|x) \sup_y d_{\infty,p}\left(\tilde{V}_1(\cdot; y), \tilde{V}_2(\cdot; y)\right)$$

$$= \gamma^p d_{\infty,p}\left(\tilde{V}_1, \tilde{V}_2\right) \underbrace{\int \mathcal{P}^\pi(\mathrm{d}y|x)}_{=1} = \gamma^p d_{\infty,p}\left(\tilde{V}_1, \tilde{V}_2\right).$$

Since this holds for any $x \in \mathcal{X}$, we get the desired result by taking the supremum over $x \in \mathcal{X}$ in the left-hand side (LHS).

The proof of the result w.r.t. $d_{1,p}$ is similar. For any $x \in \mathcal{X}$, we have

$$d_{1,p}\left((\tilde{T}^\pi \tilde{V}_1)(\cdot; x), (\tilde{T}^\pi \tilde{V}_2)(\cdot; x)\right) = \int \left|\frac{\tilde{R}(\omega; x) \int \mathcal{P}^\pi(\mathrm{d}y|x) \left(\tilde{V}_1(\gamma\omega; y) - \tilde{V}_2(\gamma\omega; y)\right)}{\omega^p}\right| \mathrm{d}\omega$$

$$\leq \int \underbrace{\left|\tilde{R}(\omega; x)\right|}_{\leq 1} \left[\int \mathcal{P}^\pi(\mathrm{d}y|x) \left|\frac{\tilde{V}_1(\gamma\omega; y) - \tilde{V}_2(\gamma\omega; y)}{\omega^p}\right|\right] \mathrm{d}\omega$$

$$\leq \int \mathcal{P}^\pi(\mathrm{d}y|x) \int \left|\frac{\tilde{V}_1(\gamma\omega; y) - \tilde{V}_2(\gamma\omega; y)}{\omega^p}\right| \mathrm{d}\omega$$

$$= \int \mathcal{P}^\pi(\mathrm{d}y|x) \int \left|\frac{\tilde{V}_1(\nu; y) - \tilde{V}_2(\nu; y)}{\left(\frac{\nu}{\gamma}\right)^p}\right| \mathrm{d}\nu(\frac{1}{\gamma})$$

$$= \gamma^{p-1} \int \mathcal{P}^\pi(\mathrm{d}y|x) \int \left|\frac{\tilde{V}_1(\nu; y) - \tilde{V}_2(\nu; y)}{\nu^p}\right| \mathrm{d}\nu$$

$$\leq \gamma^{p-1} \int \mathcal{P}^\pi(\mathrm{d}y|x) \sup_{y \in \mathcal{X}} \int \left|\frac{\tilde{V}_1(\nu; y) - \tilde{V}_2(\nu; y)}{\nu^p}\right| \mathrm{d}\nu$$

$$= \gamma^{p-1} d_{1,p}\left(\tilde{V}_1, \tilde{V}_2\right) \underbrace{\int \mathcal{P}^\pi(\mathrm{d}y|x)}_{=1} = \gamma^{p-1} d_{1,p}\left(\tilde{V}_1, \tilde{V}_2\right).$$

$$\tag{13}$$

We used the change of variable $\nu = \gamma\omega$, which entails that $\mathrm{d}\omega = \frac{1}{\gamma}\mathrm{d}\nu$. $\qquad\square$

For the contraction to be non-trivial, and avoid having a trivial inequality such as $\infty \leq \gamma^p \infty$, we require the boundedness of $d_{\infty,p}(\tilde{V}_1, \tilde{V}_2)$ or $d_{1,p}(\tilde{V}_1, \tilde{V}_2)$. This is a condition that should be verified, and as we shall soon see holds under certain conditions.

*Remark* 1. The conventional Bellman operator is a $\gamma$-contraction w.r.t. the supremum norm. Following this commonly used norm, one could similarly define a supremum-based norm for a VCF $\tilde{V}$ as

$$\left\| \tilde{V} \right\|_\infty = \sup_{x \in \mathcal{X}} \sup_{\omega \in \mathbb{R}} \tilde{V}(\omega; x).$$

The Bellman operator $\tilde{T}^\pi$, however, is not a contraction w.r.t. this norm. To see this, consider a simple MDP with $\mathcal{P}^\pi = \mathbf{I}$ (each state returns to itself) and $\tilde{R}(\omega; x) = 1$, which corresponds to the choice of $R^\pi(\mathrm{d}y|x) = \delta(y - x)$, a Dirac's delta function. Let $\gamma > 0$. For any $\tilde{V}_1, \tilde{V}_2$, we have

$$\left\| \tilde{T}^\pi \tilde{V}_1 - \tilde{T}^\pi \tilde{V}_2 \right\|_\infty = \sup_{x \in \mathcal{X}} \sup_{\omega \in \mathbb{R}} \left| \tilde{R}(\omega; x) \left( \tilde{V}_1(\gamma \omega; x) - \tilde{V}_2(\gamma \omega; x) \right) \right|$$

$$= \sup_{x \in \mathcal{X}} \sup_{\omega \in \mathbb{R}} \left| \tilde{V}_1(\gamma \omega; x) - \tilde{V}_2(\gamma \omega; x) \right|$$

$$= \sup_{x \in \mathcal{X}} \sup_{\nu \in \mathbb{R}} \left| \tilde{V}_1(\nu; x) - \tilde{V}_2(\nu; x) \right| = \left\| \tilde{V}_1 - \tilde{V}_2 \right\|_\infty.$$

Therefore, the Bellman operator $\tilde{T}^\pi$ is a non-expansion, but not a contraction, w.r.t. $\| \cdot \|_\infty$. Having the $\omega$ term in the denominator of $d_{\infty,p}$ and $d_{1,p}$ is important to get a contraction.

The importance of showing that the Bellman operator for VCF is a contraction is that we can then apply the Banach fixed point theorem (e.g., Theorem 3.2 of Hunter and Nachtergaele [2001]) to show the uniqueness of the fixed point $\tilde{V}^\pi$ (we also require the completeness of the space, which is shown for $d_{\infty,p}$). Moreover, it suggests that we can find the fixed point by iterative application of the operator. This is the path we pursue in the next section.

## 4  Characteristic value iteration

The contraction property of the Bellman operator $\tilde{T}^\pi$ (Lemma 1) suggests that we can find $\tilde{V}^\pi$ by an iterative procedure, similar to the conventional value iteration. The procedure is

$$\tilde{V}_1 \leftarrow \tilde{R},$$
$$\tilde{V}_{k+1} \leftarrow \tilde{T}^\pi \tilde{V}_k = \tilde{R} \mathcal{P}^\pi \tilde{V}_k. \qquad (k \geq 1) \tag{14}$$

We call this procedure Characteristic Value Iteration (CVI).

CVI converges under certain conditions. To see this, notice that $\tilde{V}^\pi = \tilde{T}^\pi \tilde{V}^\pi$, so for $p \geq 1$ by Lemma 1 we have

$$d_{\infty,p}(\tilde{T}^\pi \tilde{V}_k, \tilde{V}^\pi) = d_{\infty,p}(\tilde{T}^\pi \tilde{V}_k, \tilde{T}^\pi \tilde{V}^\pi) \leq \gamma^p d_{\infty,p}(\tilde{V}_k, \tilde{V}^\pi),$$

under the condition that $d_{\infty,p}(\tilde{V}_k, \tilde{V}^\pi) < \infty$. Similarly, we have $d_{1,p}(\tilde{T}^\pi \tilde{V}_k, \tilde{V}^\pi) \leq \gamma^{p-1} d_{1,p}(\tilde{V}_k, \tilde{V}^\pi)$ (for $p > 1$). By the iterative application of this upper bound, assuming that $d_{\infty,p}(\tilde{R}, \tilde{V}^\pi) < \infty$, we get that

$$d_{\infty,p}(\tilde{V}_{k+1}, \tilde{V}^\pi) \leq \gamma^p d_{\infty,p}(\tilde{V}_k, \tilde{V}^\pi) \leq \cdots \leq (\gamma^p)^k d_{\infty,p}(\tilde{V}_1, \tilde{V}^\pi) = (\gamma^p)^k d_{\infty,p}(\tilde{R}, \tilde{V}^\pi). \tag{15}$$

Likewise, assuming that $d_{1,p}(\tilde{R}, \tilde{V}^\pi) < \infty$, we obtain

$$d_{1,p}(\tilde{V}_{k+1}, \tilde{V}^\pi) \leq (\gamma^{p-1})^k d_{1,p}(\tilde{R}, \tilde{V}^\pi). \tag{16}$$

As long as $d_{\infty,p}(\tilde{R}, \tilde{V}^\pi)$ (or $d_{1,p}(\tilde{R}, \tilde{V}^\pi)$) is finite for some $p \geq 1$ ($p > 1$), CVI converges geometrically fast. Lemma 11 in Appendix B specifies the condition when the $d_{\infty,p}$ distance of two CF would be finite. For $p = 1$, it is sufficient that the immediate reward $R^\pi(x) \sim \mathcal{R}(\cdot; x)$ and the return $G^\pi(\cdot; x)$ be integrable, i.e., $\mathbb{E}\left[ |R^\pi(x)| \right], \mathbb{E}\left[ |G^\pi(\cdot; x)| \right] < \infty$ for all states $x \in \mathcal{X}$. Since we deal with discounted MDP, the integrability of $R^\pi(x)$ (uniformly over $\mathcal{X}$) entails the integrability of $G^\pi(\cdot; x)$. Therefore under very mild conditions, CVI is convergent w.r.t. $d_{\infty,1}$.

For integer valued $p \geq 2$, the condition becomes more restrictive. The first requirement is that $\mathbb{E}\left[|R^\pi(x)|^p\right]$ and $\mathbb{E}\left[|G^\pi(\cdot;x)|^p\right]$ are finite. This is not restrictive, and holds for many problems. The restrictive condition is that the first $k = 1, \ldots, p-1$ moments of the reward and the return should match, i.e., $\mathbb{E}\left[R^\pi(x)^k\right] = \mathbb{E}\left[G^\pi(x)^k\right]$ for all $x \in \mathcal{X}$. This does not seem realistic, perhaps except for $p = 2$ when problems with zero expected immediate reward for all states but with varying variance are imaginable.

One can show that the fixed point of $\tilde{T}^\pi$ is unique.

**Proposition 2.** *Consider an MDP with a discount factor $0 \leq \gamma < 1$. Consider $\mathcal{V}_B \triangleq \left\{\tilde{V} : \tilde{V} \in \mathcal{V}, d_{\infty,1}(\tilde{V}, \tilde{V}^\pi) \leq B\right\}$ for a finite $B > 0$. The Bellman operator admits a unique fixed point in $\mathcal{V}_B$, which is $\tilde{V}^\pi$. Furthermore, the CVI procedure (14) starting from $\tilde{V}_1 \in \mathcal{V}_B$ generates a sequence $(\tilde{V}_k) \subset \mathcal{V}_B$ that converges to the fixed point. If we assume that the first absolute moment of the reward distribution is uniformly finite (i.e., $\bar{r}_{max} \triangleq \sup_{x \in \mathcal{X}} \mathbb{E}\left[|R^\pi(x)|\right] < \infty$), we may choose $\tilde{V}_1 = \tilde{R}$ and set $B = \frac{2-\gamma}{1-\gamma}\bar{r}_{max}$.*

*Proof.* Proposition 10 in Appendix B shows that the metric space $(\mathcal{V}, d_{\infty,1})$ is complete. The subset $\mathcal{V}_B$ is a closed ball in $\mathcal{V}$, hence $(\mathcal{V}_{\tilde{R}}, d_{\infty,1})$ is complete.

For any $\tilde{V}_1, \tilde{V}_2 \in \mathcal{V}_B$, we have that $d_{\infty,1}(\tilde{V}_1, \tilde{V}_2) \leq d_{\infty,1}(\tilde{V}_1, \tilde{V}^\pi) + d_{\infty,1}(\tilde{V}^\pi, \tilde{V}_2) \leq 2B < \infty$. Therefore, the finiteness condition of the upper bound in Proposition 1 is satisfied, and the Bellman operator $\tilde{T}^\pi$ is a $\gamma$-contraction within $\mathcal{V}_B$.

Moreover, the application of the Bellman operator on any $\tilde{V} \in \mathcal{V}_B$ leaves it within $\mathcal{V}_B$. To see this, notice that $d_{\infty,1}(\tilde{T}^\pi\tilde{V}, \tilde{V}^\pi) = d_{\infty,1}(\tilde{T}^\pi\tilde{V}, \tilde{T}^\pi\tilde{V}^\pi) \leq \gamma d_{\infty,1}(\tilde{V}, \tilde{V}^\pi) \leq \gamma B$, which entails that $\tilde{T}^\pi\tilde{V} \in \mathcal{V}_B$. By induction, the sequence generated by the repeated application of the Bellman operator remains within $\mathcal{V}_B$.

Given the contraction property of the Bellman operator and the completeness of the space, the Banach fixed-point theorem shows that $\tilde{T}^\pi$ admits a unique fixed point within $\mathcal{V}_B$, and the fixed point is the limit of a CVI procedure starting from any $\tilde{V}_1 \in \mathcal{V}_B$.

To show that the CVI procedure with $\tilde{V}_1 = \tilde{R}$ converges to $\tilde{V}^\pi$, we should verify that $\tilde{R}$ is within $\mathcal{V}_B$ with the choice of $B = \frac{2-\gamma}{1-\gamma}\bar{r}_{max}$. Lemma 11 in Appendix B shows that $d_{\infty,1}(\tilde{R}, \tilde{V}^\pi) \leq \sup_{x \in \mathcal{X}}\{\

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

Let us first define some new notations. For a frequency-state function $\tilde{V}$ and an integer $k \geq 0$, we use $\tilde{V}^{[k]}$ to denote the same function with the difference that its frequency domain variable $\omega$ is $\gamma^k$-scaled, that is,

$$\tilde{V}^{[k]}(\omega; x) \triangleq \tilde{V}(\gamma^k \omega; x), \qquad \forall \omega \in \mathbb{R}, \forall x \in \mathcal{X}. \tag{20}$$

Notice that $\tilde{V}^{[0]}$ is the same as $\tilde{V}$. The definition of $\tilde{R}^{[k]}$ is the same.

We define a new operator $\tilde{\mathcal{U}}^{[k]} = \tilde{R}^{[k]} \mathcal{P}^\pi$, which means that for $\tilde{V}$, we have

$$\left(\tilde{\mathcal{U}}^{[k]} \tilde{V}\right)(\omega; x) = \tilde{R}^{[k]}(\omega; x) \int \mathcal{P}^\pi(\mathrm{d}y|x) \tilde{V}(\omega; y).$$

With this notation, we have

$$\begin{aligned}
(\tilde{T}^\pi \tilde{V})(\omega; x) &= \tilde{R}(\omega; x) \int \mathcal{P}^\pi(\mathrm{d}y|x) \tilde{V}(\gamma\omega; y) \\
&= \tilde{R}(\omega; x) \int \mathcal{P}^\pi(\mathrm{d}y|x) \tilde{V}^{[1]}(\omega; y) = \left(\tilde{\mathcal{U}}^{[0]} \tilde{V}^{[1]}\right)(\omega; x).
\end{aligned}$$

The presence of the $\tilde{V}^{[1]}$ term in the RHS indicates that each application of the Bellman operator $\tilde{T}^\pi$ expands the frequency domain of the function to which it is applied by a factor of $\gamma$. Our choice of notation (20) makes this more transparent, and allows us to keep track of how much the frequency domain is expanded.

We define the operator $\mathcal{L}$, which is simply the multiplication of several $\tilde{\mathcal{U}}$, with different frequency domain scaling.

$$\mathcal{L}_i = \begin{cases} \mathbf{I} & i = 0 \\ \tilde{\mathcal{U}}^{[0]} \dots \tilde{\mathcal{U}}^{[i-1]} & i \geq 1 \end{cases} \tag{21}$$

The repeated application of the Bellman operator to $\tilde{R}$ (or any other frequency-state function $\tilde{V}$) can be written as

$$(\tilde{T}^\pi)^k \tilde{R} \triangleq \underbrace{\tilde{T}^\pi \cdots \tilde{T}^\pi}_{k \text{ times}} \tilde{V} = \tilde{\mathcal{U}}^{[0]} \tilde{\mathcal{U}}[1] \dots \tilde{\mathcal{U}}^{[k-1]} \tilde{R}^{[k]} = \mathcal{L}_k \tilde{R}^{[k]}. \tag{22}$$

The next lemma is a pointwise quantification on how the errors $\tilde{\varepsilon}_k$ propagate throughout iteration of ACVI.

**Lemma 3.** *Consider the ACVI procedure* (17). *For $k \geq 0$, we have*

$$\tilde{V}_{k+1} = \mathcal{L}_k \tilde{R}^{[k]} + \sum_{i=0}^{k} \mathcal{L}_i \tilde{\varepsilon}_{k+1-i}^{[i]}.$$

*Proof.* We expand ACVI, and simplify using the introduced notations.

$$\tilde{V}_1 \leftarrow \tilde{R}^{[0]} + \tilde{\varepsilon}_1^{[0]}$$

$$
\begin{aligned}
\tilde{V}_2 \leftarrow \tilde{T}^\pi \tilde{V}_1 + \tilde{\varepsilon}_2^{[0]} \quad &= \tilde{T}^\pi \left( \tilde{R}^{[0]} + \tilde{\varepsilon}_1^{[0]} \right) + \tilde{\varepsilon}_2^{[0]} \\
&= \tilde{R}\mathcal{P}^\pi \left( \tilde{R}^{[1]} + \tilde{\varepsilon}_1^{[1]} \right) + \tilde{\varepsilon}_2^{[0]} \\
&= \tilde{\mathcal{U}}^{[0]} \tilde{R}^{[1]} + \tilde{\mathcal{U}}^{[0]} \tilde{\varepsilon}_1^{[1]} + \tilde{\varepsilon}_2^{[0]}
\end{aligned}
$$

$$
\begin{aligned}
\tilde{V}_3 \leftarrow \tilde{T}^\pi \tilde{V}_2 + \tilde{\varepsilon}_3^{[0]} \quad &= \tilde{T}^\pi \left( \tilde{\mathcal{U}}^{[0]} \tilde{R}^{[1]} + \tilde{\mathcal{U}}^{[0]} \tilde{\varepsilon}_1^{[1]} + \tilde{\varepsilon}_2^{[0]} \right) + \tilde{\varepsilon}_3^{[0]} \\
&= \tilde{R}\mathcal{P}^\pi \left( \tilde{\mathcal{U}}^{[1]} \tilde{R}^{[2]} + \tilde{\mathcal{U}}^{[1]} \tilde{\varepsilon}_1^{[2]} + \tilde{\varepsilon}_2^{[1]} \right) + \tilde{\varepsilon}_3^{[0]} \\
&= \tilde{\mathcal{U}}^{[0]} \tilde{\mathcal{U}}^{[1]} \tilde{R}^{[2]} + \tilde{\mathcal{U}}^{[0]} \tilde{\mathcal{U}}^{[1]} \tilde{\varepsilon}_1^{[2]} + \tilde{\mathcal{U}}^{[0]} \tilde{\varepsilon}_2^{[1]} + \tilde{\varepsilon}_3^{[0]}
\end{aligned}
$$

$$\vdots$$

$$
\begin{aligned}
\tilde{V}_{k+1} \leftarrow \tilde{T}^\pi \tilde{V}_k + \tilde{\varepsilon}_{k+1}^{[0]} = &\; \tilde{\mathcal{U}}^{[0]} \cdots \tilde{\mathcal{U}}^{[k-1]} \tilde{R}^{[k]} + \\
&\; \tilde{\mathcal{U}}^{[0]} \cdots \tilde{\mathcal{U}}^{[k-1]} \tilde{\varepsilon}_1^{[k]} + \tilde{\mathcal{U}}^{[0]} \cdots \tilde{\mathcal{U}}^{[k-2]} \tilde{\varepsilon}_2^{[k-1]} + \ldots + \tilde{\mathcal{U}}^{[0]} \tilde{\varepsilon}_k^{[1]} + \tilde{\varepsilon}_{k+1}^{[0]}.
\end{aligned}
$$

Substituting the notation $\mathcal{L}_i = \tilde{\mathcal{U}}^{[0]} \cdots \tilde{\mathcal{U}}^{[i-1]}$ (21) gets us to the desired result. $\qquad\square$

*Example* 1. Consider an MDP with self-returning state transition, i.e., $\mathcal{P}^\pi = \mathbf{I}$. Suppose that we run the exact CVI for $k$ iterations, that is, $\tilde{\varepsilon}_k = 0$ for all $i = 1, \ldots, k+1$. By Lemma 3, we have

$$\tilde{V}_{k+1} = \mathcal{L}_k \tilde{R}^{[k]} = \tilde{\mathcal{U}}^{[0]} \cdots \tilde{\mathcal{U}}^{[k-1]} \tilde{R}^{[k]}.$$

As $\tilde{\mathcal{U}}^{[i]} = \tilde{R}^{[i]} \mathcal{P}^\pi = \tilde{R}^{[i]} \mathbf{I}$, we have $\tilde{V}_{k+1} = \tilde{R}^{[0]} \cdots \tilde{R}^{[k]}$. For each state $x$, the CF of the computed return is $\tilde{V}_{k+1}(\omega; x) = \tilde{R}(\omega; x) \tilde{R}(\gamma\omega; x) \ldots \tilde{R}(\gamma^k \omega; x)$. By Lemma 8, given two independent random variables $R_1$ and $R_2$ and constants $a_1$ and $a_2$, their CFs satisfy $c_{a_1 R_1 + a_2 R_2}(\omega) = c_{R_1}(a_1 \omega) c_{R_2}(a_2 \omega)$. So $\tilde{V}_{k+1}(\omega; x)$ is the CF of a r.v. $R_0 + \gamma R_1 + \ldots + \gamma^k R_k$ with each $R_i \sim R^\pi(\cdot | x)$ independently. This r.v. is the $k$-step Monte Carlo approximation of the return. And running CVI for $k$ iterations computes its CF. This observation is more general and holds for arbitrary $\mathcal{P}^\pi$.

We move from pointwise quantification of the errors (Lemma 3) to computing their norm. We first state the following intermediate result.

**Lemma 4.** *For any bounded $\tilde{V} : \mathbb{R} \times \mathcal{X} \to \mathbb{R}$, and any integers $k, l \geq 0$, we have*

$$
\left\| \left( \tilde{\mathcal{U}}^{[k]} \tilde{V} \right)(\omega; \cdot) \right\|_\infty \leq \left\| \tilde{R}^{[k]}(\omega; \cdot) \right\|_\infty \left\| \tilde{V}(\omega; \cdot) \right\|_\infty,
$$

$$
\left\| \left( \tilde{\mathcal{U}}^{[k]} \tilde{\mathcal{U}}^{[l]} \tilde{V} \right)(\omega; \cdot) \right\|_\infty \leq \left\| \tilde{R}^{[k]}(\omega; \cdot) \right\|_\infty \left\| \tilde{R}^{[l]}(\omega; \cdot) \right\|_\infty \left\| \tilde{V}(\omega; \cdot) \right\|_\infty.
$$

*Proof.* For any $x \in \mathcal{X}$, $\omega \in \mathbb{R}$ and $k \geq 0$, we have

$$
\left| \left( \tilde{\mathcal{U}}^{[k]} \tilde{V} \right)(\omega; x) \right| = \left| \tilde{R}^{[k]}(\omega; x) \int \mathcal{P}^\pi(\mathrm{d}y | x) \tilde{V}(\omega; y) \right| \leq \left| \tilde{R}^{[k]}(\omega; x) \right| \left\| \tilde{V}(\omega; \cdot) \right\|_\infty.
$$

Taking the supremum over $x \in \mathcal{X}$, we get $\| (\tilde{\mathcal{U}}^{[k]} \tilde{V})(\omega; \cdot) \|_\infty \leq \| \tilde{R}^{[k]}(\omega; \cdot) \|_\infty \| \tilde{V}(\omega; \cdot) \|_\infty$.

Similarly, we have

$$
\begin{aligned}
\left| \left( \tilde{\mathcal{U}}^{[k]} \tilde{\mathcal{U}}^{[l]} \tilde{V} \right)(\omega; x) \right| &= \left| \tilde{R}^{[k]}(\omega; x) \int \mathcal{P}^\pi(\mathrm{d}y | x) \left( \tilde{R}^{[l]}(\omega; y) \int \mathcal{P}^\pi(\mathrm{d}z | y) \tilde{V}(\omega; z) \right) \right| \\
&\leq \left\| \tilde{V}(\omega; \cdot) \right\|_\infty \left| \tilde{R}^{[k]}(\omega; x) \int \mathcal{P}^\pi(\mathrm{d}y | x) \left| \tilde{R}^{[l]}(\omega; y) \right| \right| \\
&\leq \left\| \tilde{V}(\omega; \cdot) \right\|_\infty \left| \tilde{R}^{[k]}(\omega; x) \right| \left\| \tilde{R}^{[l]}(\omega; \cdot) \right\|_\infty.
\end{aligned}
$$

Taking the supremum over the state space leads to the desired result. $\qquad\square$

The following theorem is the main result of this section.

**Theorem 5.** *Consider the ACVI procedure* (17) *after $K \geq 1$ iterations. Assume that $\tilde{\varepsilon}_k(0; x) = 0$ for all $x \in \mathcal{X}$ and $k = 1, \dots, K + 1$. We have*

$$d_{\infty,p}(\tilde{V}_{K+1}, \tilde{V}^\pi) \leq \sum_{i=0}^{K} (\gamma^p)^i \|\tilde{\varepsilon}_{K+1-i}\|_{\infty,p} + (\gamma^p)^K d_{\infty,p}(\tilde{R}, \tilde{V}^\pi), \qquad (p \geq 1)$$

$$d_{1,p}(\tilde{V}_{K+1}, \tilde{V}^\pi) \leq \sum_{i=0}^{K} (\gamma^{p-1})^i \|\tilde{\varepsilon}_{K+1-i}\|_{1,p} + (\gamma^{p-1})^K d_{1,p}(\tilde{R}, \tilde{V}^\pi). \qquad (p > 1)$$

*Proof.* We decompose the error to two parts, one from stopping the exact CVI after $K$ iterations instead of letting $K \to \infty$, and the other is from only approximately performing CVI, which is encoded by having $\tilde{\varepsilon}_k \neq 0$.

We denote the CVF of applying $K$ iterations of the exact CVI by $\tilde{V}_{K+1}^\circ$. It is equal to $K$-times application $\tilde{T}^\pi$ to $\tilde{R}$, that is $\tilde{V}_{K+1}^\circ = (\tilde{T}^\pi)^{[K]} \tilde{R} = \mathcal{L}_K \tilde{R}^{[K]}$ (cf. (22)).

By the triangle inequality,

$$d_{\infty,p}(\tilde{V}_{K+1}, \tilde{V}^\pi) \leq d_{\infty,p}(\tilde{V}_{K+1}, \tilde{V}_{K+1}^\circ) + d_{\infty,p}(\tilde{V}_{K+1}^\circ, \tilde{V}^\pi),$$

$$d_{1,p}(\tilde{V}_{K+1}, \tilde{V}^\pi) \leq d_{1,p}(\tilde{V}_{K+1}, \tilde{V}_{K+1}^\circ) + d_{1,p}(\tilde{V}_{K+1}^\circ, \tilde{V}^\pi). \qquad (23)$$

By (15) and (16), we get

$$d_{\infty,p}(\tilde{V}_{K+1}^\circ, \tilde{V}^\pi) \leq (\gamma^p)^K d_{\infty,p}(\tilde{R}, \tilde{V}^\pi),$$

$$d_{1,p}(\tilde{V}_{K+1}^\circ, \tilde{V}^\pi) \leq (\gamma^{p-1})^K d_{1,p}(\tilde{R}, \tilde{V}^\pi). \qquad (24)$$

Let us attend to $d_{\infty,p}(\tilde{V}_{K+1}, \tilde{V}_{K+1}^\circ)$ and $d_{1,p}(\tilde{V}_{K+1}, \tilde{V}_{K+1}^\circ)$. Lemma 3 shows that

$$\tilde{V}_{K+1} = \mathcal{L}_K \tilde{R}^{[K]} + \sum_{i=0}^{K} \mathcal{L}_i \tilde{\varepsilon}_{K+1-i}^{[i]}.$$

As $d_{\infty,p}(\tilde{V}_{K+1}, \tilde{V}_{K+1}^\circ) = d_{\infty,p}(\tilde{V}_{K+1}, \mathcal{L}_K \tilde{R}^{[K]})$, and likewise for $d_{1,p}$, we need to provide upper bounds for

$$d_{1,p}(\tilde{V}_{K+1}, \mathcal{L}_K \tilde{R}^{[K]}) \leq d_{1,p}\left( \sum_{i=0}^{K} \mathcal{L}_i \tilde{\varepsilon}_{K+1-i}^{[i]}, 0 \right),$$

$$d_{\infty,p}(\tilde{V}_{K+1}, \mathcal{L}_K \tilde{R}^{[K]}) \leq d_{\infty,p}\left( \sum_{i=0}^{K} \mathcal{L}_i \tilde{\varepsilon}_{K+1-i}^{[i]}, 0 \right).$$

By the repeated application of Lemma 4, we have that for any $\tilde{\varepsilon}$ and $i \geq 0$,

$$\left\| \left( \mathcal{L}_i \tilde{\varepsilon}^{[i]} \right)(\omega; \cdot) \right\|_\infty = \left\| \left( \tilde{\mathcal{U}}^{[0]} \cdots \tilde{\mathcal{U}}^{[i-1]} \tilde{\varepsilon}^{[i]} \right)(\omega; \cdot) \right\|_\infty \leq \left( \prod_{j=0}^{i-1} \left\| \tilde{R}^{[j]}(\omega; \cdot) \right\|_\infty \right) \left\| \tilde{\varepsilon}^{[i]} \right\|_\infty \qquad (25)$$

We consider the case of $d_{\infty,p}$ first. Using (25) and the fact that the absolute value $|\tilde{R}(\omega;x)| \le 1$ (for all $\omega \in \mathbb{R}$) because $\tilde{R}(\cdot;x)$ is a CF (Lemma 8), we get that

$$
d_{\infty,p}\left(\sum_{i=0}^{K} \mathcal{L}_i \tilde{\varepsilon}_{K+1-i}^{[i]}, 0\right) = \sup_x \sup_\omega \left|\frac{\sum_{i=0}^{K}\left(\mathcal{L}_i \tilde{\varepsilon}_{K+1-i}^{[i]}\right)(\omega;x)}{\omega^p}\right|
$$

$$
\le \sum_{i=0}^{K} \sup_x \sup_\omega \left|\frac{\left(\mathcal{L}_i \tilde{\varepsilon}_{K+1-i}^{[i]}\right)(\omega;x)}{\omega^p}\right|
$$

$$
\le \sum_{i=0}^{K} \sup_\omega \frac{\left(\prod_{j=0}^{i-1}\left\|\tilde{R}^{[j]}(\omega;\cdot)\right\|_\infty\right)\left\|\tilde{\varepsilon}_{K+1-i}^{[i]}(\omega;\cdot)\right\|_\infty}{|\omega|^p}
$$

$$
\le \sum_{i=0}^{K} \sup_\omega \frac{\left\|\tilde{\varepsilon}_{K+1-i}^{[i]}(\omega;\cdot)\right\|_\infty}{|\omega|^p} \tag{26}
$$

For any $c : \mathbb{R} \to \mathbb{R}$ and $\gamma > 0$ (cf. (12)),

$$
\sup_{\omega \in \mathbb{R}} \left|\frac{c(\gamma^i \omega)}{\omega^p}\right| = (\gamma^p)^i \sup_{\omega \in \mathbb{R}} \left|\frac{c(\omega)}{\omega^p}\right|.
$$

This allows us to simplify (26) to

$$
d_{\infty,p}\left(\sum_{i=0}^{K} \mathcal{L}_i \tilde{\varepsilon}_{K+1-i}^{[i]}, 0\right) \le \sum_{i=0}^{K} (\gamma^p)^i \sup_\omega \frac{\|\tilde{\varepsilon}_{K+1-i}(\omega;\cdot)\|_\infty}{|\omega|^p} = \sum_{i=0}^{K} (\gamma^p)^i \|\tilde{\varepsilon}_{K+1-i}\|_{\infty,p}. \tag{27}
$$

We now consider the case of $d_{1,p}$, which is similar.

$$
d_{1,p}\left(\sum_{i=0}^{K} \mathcal{L}_i \tilde{\varepsilon}_{K+1-i}^{[i]}, 0\right) = \sup_x \int \left|\frac{\sum_{i=0}^{K}\left(\mathcal{L}_i \tilde{\varepsilon}_{K+1-i}^{[i]}\right)(\omega;x)}{\omega^p}\right| \mathrm{d}\omega
$$

$$
\le \sum_{i=0}^{K} \sup_x \int \left|\frac{\left(\mathcal{L}_i \tilde{\varepsilon}_{K+1-i}^{[i]}\right)(\omega;x)}{\omega^p}\right| \mathrm{d}\omega
$$

$$
= \sum_{i=0}^{K} \int \frac{\sup_x \left|\left(\mathcal{L}_i \tilde{\varepsilon}_{K+1-i}^{[i]}\right)(\omega;x)\right|}{|\omega|^p} \mathrm{d}\omega
$$

$$
\le \sum_{i=0}^{K} \int \left(\prod_{j=0}^{i-1}\left\|\tilde{R}^{[j]}(\omega;\cdot)\right\|_\infty\right)\frac{\left\|\tilde{\varepsilon}_{K+1-i}^{[i]}(\omega;\cdot)\right\|_\infty}{|\omega|^p} \mathrm{d}\omega
$$

$$
\le \sum_{i=0}^{K} \int \frac{\left\|\tilde{\varepsilon}_{K+1-i}^{[i]}(\omega;\cdot)\right\|_\infty}{|\omega|^p} \mathrm{d}\omega. \tag{28}
$$

By using the same change of variable used in (13) in the proof of Lemma 1, we have that for $c : \mathbb{R} \to \mathbb{R}$, $\gamma > 0$, and $p > 1$,

$$
\int \frac{|c(\gamma^i \omega)|}{|\omega|^p} \mathrm{d}\omega = \int \frac{|c(\nu)|}{|\frac{\nu}{\gamma}|^p}\frac{1}{\gamma} \mathrm{d}\nu = (\gamma^{p-1})^i \int \frac{|c(\omega)|}{|\omega|^p} \mathrm{d}\omega.
$$

This allows us to simplify (28) to

$$
d_{1,p}\left(\sum_{i=0}^{K} \mathcal{L}_i \tilde{\varepsilon}_{K+1-i}^{[i]}, 0\right) \le \sum_{i=0}^{K} (\gamma^{p-1})^i \int \frac{\|\tilde{\varepsilon}_{K+1-i}(\omega;\cdot)\|_\infty}{|\omega|^p} \mathrm{d}\omega = \sum_{i=0}^{K} (\gamma^{p-1})^i \|\tilde{\varepsilon}_{K+1-i}\|_{1,p}. \tag{29}
$$

Plugging (24), (27), and (29) in (23) leads to the final result. $\qquad\square$

This result shows how the errors $\tilde{\varepsilon}_k$ in the ACVI procedure propagate throughout iterations and affect the quality of the approximation of $\tilde{V}^\pi$ by $\tilde{V}_{K+1}$. The error is measured according to the distances $d_{1,p}$ and $d_{\infty,p}$. The upper bounds show that errors in the earlier iterations are geometrically decayed. This entails that if the resources are limited, it is better to ensure the smallness of errors in later iterations. This phenomenon is similar to what we have observed in the conventional value iteration [Farahmand et al., 2010].

As discussed in Section 4, the condition that $d_{\infty,p}(\tilde{R}, \tilde{V}^\pi)$ is finite might be very restrictive for $p > 2$ and even for $p = 2$, it might hold only in special problems. But the finiteness of $d_{\infty,1}$ requires mild conditions. For the finiteness of $d_{\infty,1}(\tilde{R}, \tilde{V}^\pi)$ in the upper bound, the finiteness of the first absolute moment of the reward function is sufficient, as discussed after (16). For the finiteness of $\|\tilde{\varepsilon}_i\|_{\infty,1}$ terms, it is sufficient that $\tilde{\varepsilon}_i(0; x) = 0$ and that its first derivative w.r.t. $\omega$ is bounded for all states $x \in \mathcal{X}$, i.e., $|\tilde{\varepsilon}^{(1)}(\omega; x)| < \infty$ (this can be seen from the proof of Lemma 11 in Appendix B). Based on these, so from now on we focus on $p = 1$.

## 6 From error in frequency domain to error in probability distributions

Theorem 5 in the previous section relates the errors at each iteration of ACVI to the quality of the obtained approximation of $\tilde{V}^\pi$. The error is measured according to the metrics $d_{1,p}$ and $d_{\infty,p}$. These are metrics in the frequency domain. What does having a small error in the frequency domain imply about the quality of approximating the distribution of returns $\bar{V}^\pi$?

From Levy's continuity theorem we know that the pointwise convergence of CF implies the convergence in distribution of their corresponding distributions. This suggest that we could define the error in the frequency domain

$$d_{\text{unif}}(\tilde{V}, \tilde{V}^\pi) = \sup_{x \in \mathcal{X}} \sup_{\omega \in \mathbb{R}} \left| \tilde{V}(\omega; x) - \tilde{V}^\pi(\omega; x) \right|.$$

Nevertheless, we did not define the distance this way because the Bellman operator would not be a contraction w.r.t. to it. So a valid question is whether, or in what sense, the smallness of $d_{\infty,p}(\tilde{V}, \tilde{V}^\pi)$ implies anything about the closeness of their corresponding probability distribution functions $\bar{V}$ and $\bar{V}^\pi$? In this section we show that such a relation indeed exists. We relate $d_{\infty,p}$ and $d_{1,p}$ to the $p$-smooth Wasserstein distance of the probability distribution functions [Arras et al., 2017].

**Definition 1.** *Let $p \geq 1$, $\mathcal{C}^p(\Omega)$ be the space of $p$-times continuous differentiable functions on domain $\Omega$, and $\mathcal{F}_p(\Omega) = \left\{ f \in \mathcal{C}^p(\Omega) : \|f^{(k)}\|_\infty \leq 1, 0 \leq k \leq p \right\}$. For two probability distributions $\mu_1, \mu_2 \in \mathcal{M}(\Omega)$, the $p$-smooth Wasserstein distance is defined as*

$$\mathcal{W}_{\mathcal{C}_p}(\mu_1, \mu_2) = \sup_{f \in \mathcal{F}_p(\Omega)} \left| \int f(x) \left( \mathrm{d}\mu_1(x) - \mathrm{d}\mu_2(x) \right) \right|.$$

*Remark* 2. Note that the conventional 1-Wasserstein distance is defined as

$$\mathcal{W}_1(\mu_1, \mu_2) = \sup_{f \in \text{Lip}_1(\Omega)} \left| \int f(x) \left( \mathrm{d}\mu_1(x) - \mathrm{d}\mu_2(x) \right) \right|,$$

where $\text{Lip}_1$ is the space of 1-Lipschitz functions. As $\|f^{(1)}\|_\infty \leq 1$ implies 1-Lipschitz functions, but not necessarily vice versa, $\mathcal{W}_{\mathcal{C}_1}(\mu_1, \mu_2) \leq \mathcal{W}_1(\mu_1, \mu_2)$.

Let us also define the $p$-smooth Wasserstein between $\bar{V}_1$ and $\bar{V}_2$ as follows:

$$\mathcal{W}_{\mathcal{C}_p}(\bar{V}_1, \bar{V}_2^\pi) \triangleq \sup_{x \in \mathcal{X}} \mathcal{W}_{\mathcal{C}_p}(\bar{V}_1(\cdot; x), \bar{V}_2^\pi(\cdot; x)).$$

This is the maximum over states $x \in \mathcal{X}$ of the value of the $p$-smooth Wasserstein between the distribution of return according to the probability distributions $\bar{V}_1(\cdot; x)$ and $\bar{V}_2(\cdot; x)$.

The following result provides an upper bound for the $p$-smooth Wasserstein distances of two r.v. based on the distance of their CFs (9) in the frequency domain.

**Lemma 6.** *Consider the domain $\Omega = [-B, B]$ with $0 < B < \infty$. Let $X_1$ and $X_2$ be two random variables with the probability distribution functions $\mu_1, \mu_2 \in \mathcal{M}(\Omega)$, and their corresponding CF*

$c_1, c_2 : \mathbb{R} \to \mathbb{C}$. *Let $p \geq 1$ be an integer. We have*

$$\mathcal{W}_{\mathcal{C}_{p+1}}(\mu_1, \mu_2) \leq \frac{2\sqrt{2B}}{\sqrt{\pi}} d_{\infty,p}(c_1, c_2),$$

$$\mathcal{W}_{\mathcal{C}_p}(\mu_1, \mu_2) \leq \frac{\sqrt{2}B}{\sqrt{\pi}} d_{1,p}(c_1, c_2).$$

*Proof.* Let $f \in \mathcal{C}_c^\infty(\mathbb{R})$ with the support in $[-B, B]$. Denote its Fourier transform $\tilde{f}$, i.e., $\tilde{f} = \frac{1}{\sqrt{2\pi}} \int f(x) e^{-j\omega x} \mathrm{d}x$. So we have $f(x) = \frac{1}{\sqrt{2\pi}} \int \tilde{f}(\omega) e^{+j\omega x} \mathrm{d}\omega$. This is a unitary convention for the Fourier transform. The difference between the expectation of $f$ w.r.t. $X_1 \sim \mu_1$ and $X_2 \sim \mu_2$ is

$$
\begin{aligned}
\mathbb{E}\left[f(X_1) - f(X_2)\right] &= \mathbb{E}\left[\frac{1}{\sqrt{2\pi}} \int \tilde{f}(\omega) \left(e^{+j\omega X_1} - e^{+j\omega X_2}\right) \mathrm{d}\omega\right] \\
&= \frac{1}{\sqrt{2\pi}} \int \tilde{f}(\omega) \left(\mathbb{E}\left[e^{j\omega X_1}\right] - \mathbb{E}\left[e^{j\omega X_2}\right]\right) \mathrm{d}\omega \\
&= \frac{1}{\sqrt{2\pi}} \int \tilde{f}(\omega) \left(c_{X_1}(\omega) - c_{X_1}(\omega)\right) \mathrm{d}\omega \\
&= \frac{1}{\sqrt{2\pi}} \int \tilde{f}(\omega) \omega^p \frac{c_{X_1}(\omega) - c_{X_1}(\omega)}{\omega^p} \mathrm{d}\omega \\
&\leq \frac{1}{\sqrt{2\pi}} \int \left|\frac{c_{X_1}(\omega) - c_{X_1}(\omega)}{\omega^p}\right| |\omega|^p |\tilde{f}(\omega)| \mathrm{d}\omega, \quad (30)
\end{aligned}
$$

where we used the definition of a CF. We consider the two parts of the result separately.

**Part I)** $d_{\infty,p}(c_1, c_2)$**:** We can upper bound (30) by

$$\mathbb{E}\left[f(X_1) - f(X_2)\right] \leq \frac{1}{\sqrt{2\pi}} d_{\infty,p}(c_1, c_2) \int |\omega|^p |\tilde{f}(\omega)| \mathrm{d}\omega. \quad (31)$$

The integral can be upper bound by using the Cauchy-Schwarz inequality:

$$
\begin{aligned}
\int |\omega|^p |\tilde{f}(\omega)| \mathrm{d}\omega &= \int |\omega|^p \frac{1 + |\omega|}{1 + |\omega|} |\tilde{f}(\omega)| \mathrm{d}\omega \\
&\leq \sqrt{\int |\omega|^{2p}(1 + |\omega|)^2 |\tilde{f}(\omega)|^2 \mathrm{d}\omega} \sqrt{\int \frac{1}{(1 + |\omega|)^2} \mathrm{d}\omega} \\
&\leq 2\sqrt{\int \left(|\omega|^{2p} + |\omega|^{2p+2}\right) |\tilde{f}(\omega)|^2 \mathrm{d}\omega}. \quad (32)
\end{aligned}
$$

Here we used $\int_{-\infty}^{\infty} \frac{1}{(1+|\omega|)^2} \mathrm{d}\omega = 2$ and $(1 + |\omega|)^2 \leq 2(1 + |\omega|^2)$.

The Fourier transform of the $k$-th derivative of a function satisfies $\mathcal{F}\{f^{(k)}\} = (j\omega)^k \tilde{f}(\omega)$. So by Parseval's theorem, we have

$$\int \left|\omega^p \tilde{f}(\omega)\right|^2 + \left|\omega^{p+1} \tilde{f}(\omega)\right|^2 \mathrm{d}\omega = \int \left|f^{(p)}(x)\right|^2 + \left|f^{(p+1)}(x)\right|^2 \mathrm{d}x.$$

As the support of $f$ is $[-B, +B]$, we have that

$$
\begin{aligned}
\int \left|f^{(p)}(x)\right|^2 + \left|f^{(p+1)}(x)\right|^2 \mathrm{d}x &= \int_{-B}^{+B} \left|f^{(p)}(x)\right|^2 + \left|f^{(p+1)}(x)\right|^2 \mathrm{d}x \\
&\leq (2B) \left[\left\|f^{(p)}\right\|_\infty^2 + \left\|f^{(p+1)}\right\|_\infty^2\right].
\end{aligned}
$$

Now for any $f \in \mathcal{F}_{p+1}([-B, +B])$, the value of $\|f^{(p)}\|_\infty$ and $\|f^{(p+1)}\|_\infty$ are both less than or equal to 1, and therefore the integral (32) is upper bounded by $2\sqrt{4B}$. By combining this with (31), we get

$$\mathcal{W}_{\mathcal{C}_{p+1}}(\mu_1, \mu_2) = \sup_{f \in \mathcal{F}_{p+1}([-B, +B])} \mathbb{E}\left[f(X_1) - f(X_2)\right] \leq \frac{2\sqrt{2B}}{\sqrt{\pi}} d_{\infty,p}(c_1, c_2).$$

**Part 2)** $d_{1,p}(c_1, c_2)$**:** We can upper bound (30) by

$$\mathbb{E}\left[f(X_1) - f(X_2)\right] \leq \frac{1}{\sqrt{2\pi}} \sup_\omega \left|\omega^p \tilde{f}(\omega)\right| d_{1,p}(c_1, c_2). \tag{33}$$

Observe that for any integrable function $g : \mathbb{R} \to \mathbb{R}$ and its corresponding Fourier transform $\tilde{g}$, for any $\omega$ we have that

$$|\tilde{g}(\omega)| = \left|\int g(x)e^{-j\omega x}\mathrm{d}x\right| \leq \int \left|g(x)e^{-j\omega x}\right|\mathrm{d}x \leq \int |g(x)|\mathrm{d}x.$$

So as $(j\omega)^p \tilde{f}(\omega)$ is the Fourier transform of $f^{(p)}$, we get that

$$\sup_\omega \left|\omega^p \tilde{f}(\omega)\right| \leq \int \left|f^{(p)}(x)\right|\mathrm{d}x \leq 2B \left\|f^{(p)}\right\|_\infty,$$

where we used the boundedness of the support of $f$.

By combining this with (33), we get

$$\mathcal{W}_{\mathcal{C}_p}(\mu_1, \mu_2) = \sup_{f \in \mathcal{F}_p([-B,+B])} \mathbb{E}\left[f(X_1) - f(X_2)\right] \leq \frac{\sqrt{2}B}{\sqrt{\pi}} d_{1,p}(c_1, c_2).$$

$\square$

The proof of this theorem closely follows the same line of argument as the proof of Theorem 1 by Arras et al. [2017]. Our result is both a simplification and an extension. It is a simplification because it considers a bounded support of random variables, whereas Arras et al. [2017] allows an unbounded, but decaying, tail. The result on $d_{1,p}$ is an extension, as Arras et al. [2017] only consider $d_{\infty,p}$.

Given this result, we can use it alongside Theorem 5 to prove the main result of this section.

**Theorem 7.** *Consider the ACVI procedure (17) after $K \geq 1$ iterations. Assume that $\tilde{\varepsilon}_k(0; x) = 0$ for all $x \in \mathcal{X}$ and $k = 1, \ldots, K+1$. Furthermore, assume that the immediate reward distribution $\mathcal{R}^\pi(\cdot|x)$ is $R_{max}$-bounded. We then have*

$$\mathcal{W}_{\mathcal{C}_2}(\bar{V}_{K+1}, \bar{V}^\pi) \leq \frac{2\sqrt{2}}{\sqrt{\pi}} \sqrt{\frac{R_{max}}{1 - \gamma}} \left[\sum_{i=0}^{K} \gamma^i \left\|\tilde{\varepsilon}_{K+1-i}\right\|_{\infty,1} + \frac{2\gamma^K}{1 - \gamma}R_{max}\right].$$

*Proof.* We evoke Theorem 5 with the choice of $p = 1$ to upper bound $d_{\infty,1}(\tilde{V}_{K+1}, \tilde{V}^\pi)$. This in turn provides a pointwise (over states $x \in \mathcal{X}$) upper bound guarantee for $d_{\infty,1}(\tilde{V}_{K+1}(\cdot; x), \tilde{V}^\pi(\cdot; x))$. This allows us to focus on the CF of each state separately. So Lemma 6 shows that for each $x \in \mathcal{X}$,

$$\mathcal{W}_{\mathcal{C}_2}(\bar{V}_{K+1}(\cdot; x), \bar{V}^\pi(\cdot; x)) \leq \frac{2\sqrt{2}}{\sqrt{\pi}} \sqrt{\frac{R_{max}}{1 - \gamma}} \left[\sum_{i=0}^{K} \gamma^i \left\|\tilde{\varepsilon}_{K+1-i}\right\|_{\infty,1} + \gamma^K d_{\infty,1}(\tilde{R}(\cdot; x), \tilde{V}^\pi(\cdot; x))\right].$$

It remains to upper bound $d_{\infty,1}(\tilde{R}(\cdot; x), \tilde{V}^\pi(\cdot; x))$. Because the immediate reward distribution is $R_{max}$-bounded, the distribution of random returns $\bar{V}^\pi$ would be $\frac{R_{max}}{1-\gamma}$-bounded. Lemma 11 allows us to upper bound $d_{\infty,1}(\tilde{R}(\cdot; x), \tilde{V}^\pi(\cdot; x))$ by $\mathbb{E}\left[|R^\pi(x)|\right] + \mathbb{E}\left[|G^\pi(x)|\right] \leq R_{max} + \frac{R_{max}}{1-\gamma} = \frac{2-\gamma}{1-\gamma}R_{max} \leq \frac{2}{1-\gamma}R_{max}$. This finishes the proof. $\square$

This upper bound can be simplified if we are willing to provide a uniform over iterations upper bound on $\left\|\tilde{\varepsilon}_{K+1-i}\right\|_{\infty,1}$. In that case, we have

$$\mathcal{W}_{\mathcal{C}_2}(\bar{V}_{K+1}, \bar{V}^\pi) \leq \frac{2\sqrt{2R_{max}}}{\sqrt{\pi}(1-\gamma)^{3/2}} \left[\max_{i=1,\ldots,K+1} \left\|\tilde{\varepsilon}_i\right\|_{\infty,1} + 2\gamma^K R_{max}\right].$$

We note that the 2-smooth Wasserstein distance $\mathcal{W}_{\mathcal{C}_2}$, which is an integral probability metric [Müller, 1997], is only one of the many distances between probability distributions [Gibbs and Su, 2002]. The choice of the right probability distance most likely depends on the performance measure we would like the policy to optimize. Studying this further is an interesting topic of future research.

# 7 Conclusion

This paper laid the groundwork for a new class of distributional RL algorithms. We have shown that one might represent the uncertainty about the return in the frequency domain, and such a representation (called Characteristic Value Function) enjoys properties such as satisfying a Bellman equation and having a contractive Bellman operator. This in turn allows us to compute the CVF by an iterative method called Characteristic Value Iteration. We also showed the effect of errors in the iterative procedure, and provided error propagation results, in both the frequency domain and the probability distribution space.

This paper is only the first step towards understanding CVFs and their properties. Among remaining questions is how to perform the regression step (18) of ACVI properly and efficiently. Specifically, how should we set the weighting function $w(\omega)$ in order to achieve accurate CVF in frequencies that are relevant for the tasks we want to solve. Studying other distances between CFs and their properties is another interesting research directions. This work only focused on the policy evaluation problem, so another obvious direction is designing risk-aware policy optimization algorithms based on CVF. Finally, empirically evaluating this approach for return uncertainty representation may lead to better understanding of its strengths and weaknesses.

# A    Characteristic function of a random variable

Given a real-valued random variable $X$ with the probability distribution $\mu \in \mathcal{M}(\mathbb{R})$, the space of probability distributions over $\mathbb{R}$, its corresponding CF $c_X : \mathbb{R} \to \mathbb{C}$ is the function defined as

$$c_X(\omega) \triangleq \mathbb{E}\left[e^{jX\omega}\right] = \int \exp(jx\omega)\mu(\mathrm{d}x), \qquad \omega \in \mathbb{R}$$

where $j = \sqrt{-1}$ is the imaginary unit. If the distribution has a density $p(x) = \frac{\mathrm{d}\mu}{\mathrm{d}\lambda}$ w.r.t. the Lebesgue measure $\lambda$, we have $c_X(\omega) = \int \exp(jx\omega)p(x)\mathrm{d}x$ too.

The CF of a probability distribution is closely related to the Fourier transform of its probability distribution function. The Fourier transform of a function $f : \mathbb{R} \to \mathbb{C}$ is defined as[5]

$$\tilde{f}(\omega) \triangleq \mathcal{F}\{f\}(\omega) = \int f(x)e^{-j\omega x}\mathrm{d}x.$$

Hence, $c_X$ is $\overline{\mathcal{F}\{p\}}$, the complex conjugate of the Fourier transform of the density $p$.

If $X$ has a probability distribution $\mu_\theta$ parameterized by $\theta$, we may refer to its CF by $c_\theta$.

Given independent samples $X_1, \ldots, X_n$ from $\mu$, the Empirical Characteristic Function (ECF) is defined as

$$c_n(w) \triangleq \frac{1}{n}\sum_{i=1}^{n} e^{jX_i\omega}. \qquad \forall \omega \in \mathbb{R}$$

ECF can be seen as the CF of the empirical measure, which assigns the probability

$$\mu_n(A) = \frac{1}{n}\sum_{i=1}^{n} \mathbb{I}\{X_i \in A\},$$

to any measurable set $A$ of $\mathbb{R}$ (of an appropriate $\sigma$-algebra, e.g., Borel $\sigma$-algebra). It is easy to see that because of the law of large numbers, $c_n(\omega) \to c_X(\omega)$ (a.s.) for any fixed $\omega$.

We collect some useful properties of the CF in the following lemma, see e.g., Chapters 16 and 18 of Williams [1991] or Chapter 11 of Rosenthal [2006].

**Lemma 8.** *The characteristic function of a random variable $X$ has the following properties:*

- $c_X(0) = 1$.

- $|c_X(\omega)| \le 1$ *for all* $\omega \in \mathbb{R}$.

- *The function $\omega \mapsto c_X(\omega)$ is uniformly continuous in $\mathbb{R}$.*

- $c_{(-X)}(\omega) = \overline{c_X(\omega)}.$

- $c_{aX+b}(\omega) = e^{jb\omega}c_X(a\omega).$

- *If $X$ and $Y$ are two (conditionally) independent random variables, $c_{X+Y}(\omega) = c_X(\omega)c_Y(\omega).$*

- *If $k \in \mathbb{N}$ and $\mathbb{E}\left[|X|^k\right] < \infty$, the function $c_X(\omega)$ is $k$ times differentiable and we have $c_X^{(k)}(\omega) = \mathbb{E}\left[(jX)^k e^{jX\omega}\right]$. In particular, the $k$-th moment of $X$ satisfies $c_X^{(k)}(0) = j^k \mathbb{E}\left[X^k\right].$*

- *(Levy Inversion Formula) If $\int |c_X(\omega)|\mathrm{d}\omega < \infty$, then $X$ has a continuous probability density function $p(x)$ and $p(x) = \frac{1}{2\pi} \int \exp(-j\omega x)c_X(\omega)\mathrm{d}\omega.$*

- *(Levy's Convergence Theorem) Let $(\mu_n)$ be a sequence of probability distributions, and let $(c_n)$ denote their corresponding CF. Suppose that $c(\omega) = \lim c_n(\omega)$ exists for all $\omega \in \mathbb{R}$ and $c$ is continuous at $0$. Then $c$ is a CF of some distribution $\mu$ and $\mu_n \to \mu$ in distribution.*

Note that we stated a simplified Levy Inversion formula; a more general inversion formula exists even when the r.v. $X$ does not have a density.

## B  Properties of the distance metrics $d_{1,p}$ and $d_{\infty,p}$

We provide some properties of the distances $d_{\infty,p}$ and $d_{1,p}$.

**Proposition 9.** *The distance functions $d_{1,p}$ and $d_{\infty,p}$ are metrics.*

*Proof.* We first consider $d_{\infty,p}$ defined for the CF and verify the properties of being a metric. The verification for $d_{\infty,p}$ for $\tilde{V}$ is similar.

It is clear that $d_{\infty,p}(c_1, c_2) \geq 0$. Whenever $d_{\infty,p}(c_1, c_2) = \sup_\omega \left|\frac{c_1(\omega)-c_2(\omega)}{\omega^p}\right|$ is equal to zero, it entails that $c_1(\omega) = c_2(\omega)$ for all $\omega \in \mathbb{R}$.

We also have $d_{\infty,p}(c_1, c_2) = d_{\infty,p}(c_2, c_1)$.

Finally, notice that for $c_1, c_2, c_3 : \mathbb{R} \to \mathbb{C}$, we have

$$
\begin{aligned}
d_{\infty,p}(c_1, c_2) = \sup_\omega \left|\frac{c_1(\omega) - c_2(\omega)}{\omega^p}\right| &= \sup_\omega \left|\frac{c_1(\omega) - c_3(\omega) + c_3(\omega) - c_2(\omega)}{\omega^p}\right| \\
&\leq \sup_\omega \left[\frac{|c_1(\omega) - c_3(\omega)| + |c_3(\omega) - c_2(\omega)|}{|\omega|^p}\right] \\
&\leq \sup_\omega \left|\frac{c_1(\omega) - c_3(\omega)}{\omega^p}\right| + \sup_\omega \left|\frac{c_3(\omega) - c_2(\omega)}{\omega^p}\right| \\
&= d_{\infty,p}(c_1, c_3) + d_{\infty,p}(c_3, c_2)
\end{aligned}
$$

The proof for $d_{1,p}$ is the same except that $\int \left|\frac{c_1(\omega)-c_2(\omega)}{\omega^p}\right| \mathrm{d}\omega = 0$ only entails that $c_1(\omega) = c_2(\omega)$ almost surely. $\square$

We define the space of VCF $\mathcal{V} = \left\{ \tilde{V} : \mathbb{R} \times \mathcal{X} \to \mathbb{C}_1 \; : \; \tilde{V}(0; x) = 1 \right\}$.[6] We want to show that $\mathcal{V}$ with metric $d_{\infty,p}$ is a complete space. Showing that a space is complete allows us to use Banach fixed point theorem to show that the fixed point of a contraction operator is within the space; if the space is not complete, the fixed point might be outside the space.

**Proposition 10.** *The metric space $(\mathcal{V}, d_{\infty,p})$ is complete.*

*Proof.* Let $(\tilde{V}_n)$ be a Cauchy sequence in $\mathcal{V}$ w.r.t. $d_{\infty,p}$. To show that $\mathcal{V}$ is complete, we have to prove that there exists a $\tilde{V} \in \mathcal{V}$ such that $d_{\infty,p}(\tilde{V}_n, \tilde{V}) \to 0$ as $n \to \infty$.[7]

The fact that $(\tilde{V}_n)$ is a Cauchy sequence in $d_{\infty,p}$ means that for any $\varepsilon > 0$, there exists an integer $N$ such that for any $n, m \geq N$, we have $d_{\infty,p}(\tilde{V}_n, \tilde{V}_m) < \varepsilon$. So

$$\sup_{x \in \mathcal{X}} \sup_{\omega} \left| \frac{\tilde{V}_n(\omega; x) - \tilde{V}_m(\omega; x)}{\omega^p} \right| < \varepsilon \Rightarrow \left| \tilde{V}_n(\omega; x) - \tilde{V}_m(\omega; x) \right| < \varepsilon |\omega|^p, \quad \forall \omega \in \mathbb{R} \backslash \{0\}, \forall x \in \mathcal{X}.$$

We would like to show that for any fix $x \in \mathcal{X}$ and $\omega \neq 0$, the sequence $(\tilde{V}_n(\omega; x))$ is Cauchy too. For any $\varepsilon' > 0$, let us pick $\varepsilon = \frac{\varepsilon'}{|\omega|^p}$. As $(\tilde{V}_n)$ is Cauchy w.r.t. $d_{\infty,p}$, there exists an integer number $N$ such that for any $n, m \geq N$, we have $\left| \frac{\tilde{V}_n(\omega; x) - \tilde{V}_m(\omega; x)}{\omega^p} \right| < \varepsilon$. This is equivalent to $\left| \tilde{V}_n(\omega; x) - \tilde{V}_m(\omega; x) \right| < \varepsilon |\omega|^p = \varepsilon'$, which shows that $(\tilde{V}_n(\omega; x))$ is indeed a Cauchy sequence.

As the real-valued sequence $(\tilde{V}_n(\omega; x))$ is a Cauchy sequence and $\mathbb{R}$ is complete, the pointwise sequence $\tilde{V}_n(\omega; x)$ converges to a limit. We define the following function for all $x \in \mathcal{X}$ and $\omega \in \mathbb{R} \backslash \{0\}$.

$$\tilde{V}(\omega; x) = \lim_{n \to \infty} \tilde{V}_n(\omega; x)$$

For $\omega = 0$, we pick $\tilde{V}(0; x) = 1$.

As $\tilde{V}(\omega; x) \to \tilde{V}_m(\omega; x)$ when $m \to \infty$, we have

$$d_{\infty,p}(\tilde{V}_n, \tilde{V}) = \sup_{x,\omega} \left| \frac{\tilde{V}_n(\omega; x) - \tilde{V}(\omega; x)}{\omega^p} \right|$$

$$= \sup_{x,\omega} \lim_{m \to \infty} \left| \frac{\tilde{V}_n(\omega; x) - \tilde{V}_m(\omega; x)}{\omega^p} \right|$$

$$\leq \liminf_{m \to \infty} \sup_{x,\omega} \left| \frac{\tilde{V}_n(\omega; x) - \tilde{V}_m(\omega; x)}{\omega^p} \right| = \liminf_{m \to \infty} d_{\infty,p}(\tilde{V}_n, \tilde{V}_m).$$

As $(\tilde{V}_n)$ is a Cauchy sequence, for all $\varepsilon > 0$, there exists $N$ such that for all $n, m \geq N$, $d_{\infty,p}(\tilde{V}_n, \tilde{V}_m) < \varepsilon$, which along the inequality above show that $d_{\infty,p}(\tilde{V}_n, \tilde{V}) < \varepsilon$ for all $n \geq N$. This proves that $\lim_{n \to \infty} d_{\infty,p}(\tilde{V}_n, \tilde{V}) = 0$, as desired. □

Showing whether $d_{1,p}$ also defines a complete metric space is an interesting question postponed to a future work.

We provide a condition under which the distance $d_{\infty,p}$ between two CF $c_1, c_2$ would be finite.

**Lemma 11.** *Consider two random variables $X_1$ and $X_2$ with their corresponding CF $c_1, c_2 : \mathbb{R} \to \mathbb{C}$. Let $p \geq 1$. If*

1. *(Matched first $p - 2$-th moments) $\mathbb{E}\left[X_1^k\right] = \mathbb{E}\left[X_2^k\right]$ for $k = 1, \ldots, p - 1$ (for $p \geq 2$),*

2. *(Finite $p$-th moments) $\mathbb{E}\left[|X_1|^p\right], \mathbb{E}\left[|X_2|^p\right] < \infty$,*

*the distance $d_{\infty,p}(c_1, c_2)$ would be finite and can be upper bounded by $\frac{\mathbb{E}[|X_1|^p] + \mathbb{E}[|X_2|^p]}{p!}$.*

*Proof.* We use Taylor series expansion of $c_1$ and $c_2$ to provide an upper bound on $\frac{c_1(\omega) - c_2(\omega)}{\omega^p}$. As $\mathbb{E}\left[|X|^p\right]$ is finite for both $X_1$ and $X_2$, they are finite for $k = 0, \ldots, p - 1$ too. By Lemma 8, the functions $c_1(\omega)$ and $c_2(\omega)$ are differentiable for $k = 0, \ldots, p$, and for both of them we have

$$c(\omega) = c(0) + c^{(1)}(0) \frac{\omega^1}{1!} + \ldots + c^{(p-1)}(0) \frac{\omega^{p-1}}{(p-1)!} + c^{(p)}(\nu) \Big|_{0 \leq \nu \leq \omega} \frac{\omega^p}{p!}.$$

Therefore for any $\omega$, we can write

$$\frac{c_1(\omega) - c_2(\omega)}{\omega^p} = \sum_{k=0}^{p-1} \left( c_1^{(k)}(0) - c_1^{(k)}(0) \right) \frac{1}{k! \omega^{p-k}} + \frac{1}{p!} \left( c_1^{(p)}(\nu_1) \Big|_{0 < \nu_1 < \omega} - c_2^{(p)}(\nu_1) \Big|_{0 < \nu_1 < \omega} \right).$$

Note that if $c_1^{(k)}(0) \neq c_2^{(k)}(0)$ (for $k = 0, \dots, p-1$), the corresponding term in the summation would be singular at $\omega = 0$. Because $c_X^{(k)}(0) = j^k \mathbb{E}\left[X^k\right]$, the condition of moments of the random variables being matched implies that the summation is zero. Under that condition, we have

$$\frac{c_1(\omega) - c_2(\omega)}{\omega^p} = \frac{c_1^{(p)}(\nu_1) - c_2^{(p)}(\nu_2)}{p!},$$

for some $0 \leq \nu_1, \nu_2 \leq \omega$. We use the definition of CF to write

$$\begin{aligned}
\left| c_1^{(p)}(\nu_1) - c_2^{(p)}(\nu_2) \right| &= \left| \mathbb{E}\left[ (jX_1)^p e^{j\nu_1 X_1} \right] - \mathbb{E}\left[ (jX_2)^p e^{j\nu_2 X_2} \right] \right| \\
&\leq \mathbb{E}\left[ \left| (jX_1)^p e^{j\nu_1 X_1} \right| \right] + \mathbb{E}\left[ \left| (jX_2)^p e^{j\nu_2 X_2} \right| \right] \\
&\leq \mathbb{E}\left[ |X_1|^p \right] + \mathbb{E}\left[ |X_2|^p \right].
\end{aligned}$$

Therefore, if in addition to the first condition, the $p$-th moments $\mathbb{E}\left[|X_1|^p\right]$ and $\mathbb{E}\left[|X_2|^p\right]$ are finite too,

$$d_{\infty,p}(c_1, c_2) = \sup_{\omega} \left| \frac{c_1(\omega) - c_2(\omega)}{\omega^p} \right| \leq \frac{\mathbb{E}\left[|X_1|^p\right] + \mathbb{E}\left[|X_2|^p\right]}{p!}$$

is finite too, which is the desired result. □

Observe that $d_{\infty,1}(c_1, c_2)$ is bounded as long as $\mathbb{E}\left[|X_1|\right]$ and $\mathbb{E}\left[|X_2|\right]$ are finite, which is quite mild. A similar result holds if we replace CF with CVF, i.e., $d_{\infty,p}(\tilde{V}_1, \tilde{V}_2)$. The required condition would then be on the moments of the set of random variables, indexed by $x \in \mathcal{X}$, that have CF of $\tilde{V}_1(\cdot; x)$ and $\tilde{V}_2(\cdot; x)$.

We leave the result specifying the conditions for $d_{1,p}(c_1, c_2)$ to be bounded as a future work.

## C  A Study on function approximation error

We argued in Section 4.1 that performing CVI exactly may not be practical and we have to perform it approximately. This might be done by solving a sequence of regression problems, which find $\tilde{V}_{k+1} \approx \tilde{T}^\pi \tilde{V}_k$. A regression estimator such as (18) finds the estimate within a function space $\mathcal{F}$, which is often smaller than the space of all possible CVF, which is a subset of $\mathcal{V} = \{\tilde{V} : \mathbb{R} \times \mathcal{X} \to \mathbb{C}_1 : \tilde{V}(0; x) = 1\}$. As a result, we might have some function approximation error. We would like to know whether it is possible to have small function approximation error under some reasonable assumptions on the reward distribution and the choice of $\mathcal{F}$. This section attends to this question.

The function approximation error, however, is only one source of error in the analysis of a regression algorithm. Another source of error is the estimation error, which reflects the effect of having a finite number of samples. The estimation error depends on the complexity of the function space, which can be quantified in terms of its covering number. We provide a covering number result for a particular choice of function space in Appendix D. We should note that even though studying the function approximation error and covering number of a function space are crucial steps in the error analysis of a regression method, we do not provide a complete analysis of the regression problem that should be solved at each step of CVI in this work.

Before going into the detail, we briefly describe the result: If the reward distribution is smooth in a certain sense, a band-limited function class $\mathcal{F}_b$, to be defined shortly, provides an approximation error that goes to zero as the bandwidth $b$ increases. Furthermore, if the first $s$ absolute moments of the reward distribution are finite (uniformly for all $x \in \mathcal{X}$), the CVF $\tilde{V}(\cdot; x)$ belongs to the smoothness class $C^s([-b, b]) \cap \mathcal{F}_b$. This leads to a well-behaving covering number, which can be used to obtain a convergence rate for estimation error.

Let $\mathcal{F}_b \subset \mathcal{V}$ be defined as the $b$-band-limited CVF, i.e.,

$$\mathcal{F}_b = \left\{ \tilde{V} : \mathbb{R} \times \mathcal{X} \to \mathbb{C}_1 \ : \ \tilde{V}(0; x) = 1, \tilde{V}(\omega; x) = 0 \ \forall |\omega| > b \right\}. \tag{34}$$

These are functions whose frequency component can be non-zero only in $|\omega| \leq b$. Soon we show that this function space can approximate the CF of a large class of distributions.

We define the class of $\beta$-smooth and super-smooth reward distributions, following the definition by Fan [1991].

**Definition 2.** *The reward distribution $\mathcal{R}^\pi$ is $\beta$-smooth if for all $x \in \mathcal{X}$,*

$$c_0 |\omega|^{-\beta} \leq |\tilde{R}(\omega; x)| \leq c_1 |\omega|^{-\beta},$$

*for some $c_0, c_1, \beta > 0$ and for $|\omega| \geq \omega_0$ with some finite $\omega_0 \geq 0$.*

*The reward distribution is $\beta$-super smooth if for all $x \in \mathcal{X}$,*

$$c_0 |\omega|^{\beta_0} \exp\left(-\frac{|\omega|^\beta}{\tau}\right) \leq |\tilde{R}(\omega; x)| \leq c_1 |\omega|^{\beta_1} \exp\left(-\frac{|\omega|^\beta}{\tau}\right),$$

*for some positive constants $c_0, c_1, \tau, \beta$ and constants $\beta_0$ and $\beta_1$, and for $|\omega| \geq \omega_0$ with some finite $\omega_0 \geq 0$.*

The condition of being $\beta$-smooth is satisfied by many distributions such as exponential ($\beta = 1$), uniform ($\beta = 1$), gamma ($\beta = k$, the shape parameter), etc. The condition of being super-smooth is satisfied for distributions such as normal ($\beta = 2$), Cauchy ($\beta = 1$), etc.

We would like to know that if the reward distribution is smooth (or super-smooth), what is the smallest $\|\tilde{\varepsilon}\|_{\infty,p}$ for $\tilde{\varepsilon} = \tilde{V} - \tilde{T}^\pi \tilde{V}'$ with $\tilde{V}$ being restricted to be in $\mathcal{F}_b$. The following proposition answers this question.

**Theorem 12.** *Consider function space $\mathcal{F}_b$ with $b \geq \omega_0$ (cf. Definition 2). If $\mathcal{R}$ is a $\beta$-smooth distribution, we have*

$$\sup_{\tilde{V}' \in \mathcal{V}} \inf_{\tilde{V} \in \mathcal{F}_b} \left\| \tilde{V} - \tilde{T}^\pi \tilde{V}' \right\|_{\infty,p} \leq \frac{c_1}{b^{p+\beta}},$$

$$\inf_{\tilde{V} \in \mathcal{F}_b} \left\| \tilde{V} - \tilde{R} \right\|_{\infty,p} \leq \frac{c_1}{b^{p+\beta}}.$$

*If $\mathcal{R}$ is $\beta$-super smooth distribution, under the condition that either (1) $\beta_1 \leq p$ or (2) $\beta_1 > p$ and $b > \sqrt[\beta]{\frac{\tau(\beta_1 - p)}{\beta}}$, we have*

$$\sup_{\tilde{V}' \in \mathcal{V}} \inf_{\tilde{V} \in \mathcal{F}_b} \left\| \tilde{V} - \tilde{T}^\pi \tilde{V}' \right\|_{\infty,p} \leq c_1 |b|^{\beta_1 - p} \exp\left(-\frac{b^\beta}{\tau}\right),$$

$$\inf_{\tilde{V} \in \mathcal{F}_b} \left\| \tilde{V} - \tilde{R} \right\|_{\infty,p} \leq c_1 |b|^{\beta_1 - p} \exp\left(-\frac{b^\beta}{\tau}\right).$$

*Proof.* For any $\tilde{V}' \in \mathcal{V}$, consider $(\tilde{T}^\pi \tilde{V}')(\omega; x) = \tilde{R}(\omega; x) \int \mathcal{P}^\pi(\mathrm{d}y|x) \tilde{V}'(\gamma\omega; y)$ and define a function $\tilde{V}(\omega; x) = (\tilde{T}^\pi \tilde{V}')(\omega; x) \mathbb{I}\{\omega \in [-b, +b]\}$. This function is a CF and is zero outside $[-b, b]$, so it belongs to $\mathcal{F}_b$. So $\tilde{\varepsilon} = \tilde{V} - \tilde{T}^\pi \tilde{V}'$ is $\tilde{\varepsilon}(\omega; x) = (\tilde{T}^\pi \tilde{V}')(\omega; x) \mathbb{I}\{\omega \notin [-b, +b]\}$. As $|\tilde{V}'(\omega; x)| \leq 1$ for all $x$ and $\omega$, we have

$$|\tilde{\varepsilon}(\omega; x)| = \left| \mathbb{I}\{\omega \notin [-b, +b]\} \tilde{R}(\omega; x) \int \mathcal{P}^\pi(\mathrm{d}y|x) \tilde{V}^\pi(\gamma\omega; y) \right|$$

$$\leq \mathbb{I}\{\omega \notin [-b, +b]\} \left| \tilde{R}(\omega; x) \right| \int \mathcal{P}^\pi(\mathrm{d}y|x) \left| \tilde{V}^\pi(\gamma\omega; y) \right|$$

$$\leq \mathbb{I}\{\omega \notin [-b, +b]\} \left| \tilde{R}(\omega; x) \right|.$$

Under the $\beta$-smooth condition, the norm of $\tilde{\varepsilon}$ can be upper bounded by

$$\|\tilde{\varepsilon}\|_{\infty,p} = \sup_x \sup_{|\omega|>b} \left| \frac{\tilde{R}(\omega; x)}{\omega^p} \right| \leq \sup_{|\omega|>b} \frac{c_1}{|\omega|^{p+\beta}} = \frac{c_1}{b^{p+\beta}}. \tag{35}$$

As this holds for any $\tilde{V}' \in \mathcal{V}$, by taking the supremum over $\mathcal{V}$ we obtain the first statement for the $\beta$-smooth case. The proof of the second statement is essentially the same with the difference that we choose $\tilde{V}(\omega; x) = \tilde{R}(\omega; x)\mathbb{I}\{\omega \in [-b, +b]\} \in \mathcal{F}_b$ and compute the norm of $\tilde{\varepsilon} = \tilde{R}(\omega; x)\mathbb{I}\{\omega \notin [-b, +b]\}$.

For the $\beta$-super smooth case, the argument is similar except that instead of (35), we have

$$\|\tilde{\varepsilon}\|_{\infty, p} \leq \sup_{|\omega| > b} \frac{c_1 \exp\left(-\frac{|\omega|^\beta}{\tau}\right)}{|\omega|^{p-\beta_1}}. \tag{36}$$

The function in the upper bound is not monotonically non-increasing as a function of $\omega$ (it might increase for small $\omega$ before the exponential term dominates), but under the conditions specified in the statement of the theorem, it is. In that case, we simply replace $\omega$ with $b$. $\qquad\square$

This result shows that for the class of $\beta$-smooth or super smooth reward distributions, the error in approximating $\tilde{T}^\pi \tilde{V}'$ (for any $\tilde{V}' \in \mathcal{V}$) decreases as the bandwidth $b$ increases. The rate depends on $\beta$ and $p$ for the smooth distributions as well as $\tau$ and $\beta_1$ for the super smooth ones. As discussed in Section 5, the choice of $p > 2$ is restrictive. So we summarize the result by stating that for $p = 1$, which we often care about, we have

$$\sup_{\tilde{V}' \in \mathcal{V}} \inf_{\tilde{V} \in \mathcal{F}_b} \left\|\tilde{V} - \tilde{T}^\pi \tilde{V}'\right\|_{\infty, 1} \leq \begin{cases} \frac{c_1}{b^{1+\beta}}, & \text{(smooth)} \\ c_1 |b|^{\beta_1 - 1} \exp\left(-\frac{b^\beta}{\tau}\right). & \text{(super smooth)} \end{cases}$$

Analyzing the approximation error is only one part of the error analysis of a regression estimator. Another part is the analysis of the estimation error. One may use any universally consistent regression estimator, such as a K-NN estimator or many other partitioning-based estimator, to show that the estimation error goes to zero as the number of samples increases. This along with the above approximation error lead to a controlled asymptotic upper bound.

Providing a convergence rate for the estimation error, however, requires some more (mild) assumptions on the complexity of the function class. The function class $\mathcal{F}_b$ (34) is quite large. Even if we fix a single state $x$, the function $\tilde{V}(\cdot; x)$ with $\tilde{V} \in \mathcal{F}_b$ belongs to the space of 1-bounded functions on the domain $[-b, b]$. Learning such a function can be arbitrary slow, cf. Theorem 3.1 of Györfi et al. [2002]. This might appear hopeless, but it is not.

First of all, the function space $\mathcal{F}_b$ was chosen needlessly large. A CF is uniformly continuous, so we could choose to work with the smaller space of 1-bounded continuous functions. Moreover, with some extra mild assumptions, we can define a function space that is reasonably small and can potentially lead to a convergence rate for the estimation error. In particular, if the reward distribution has $s$ finite absolute moments, its CF $\tilde{R}(\cdot; x)$ is $s$-times differentiable (Lemma 8). The space of $s$-times differentiable function is regular enough for a relatively fast convergence rate (depending on $s$). We provide the covering number result for such a space in Appendix D. In the rest of this section, we show that choosing this smaller function space still leads to reasonable function approximation properties. Let us introduce the necessary definitions.

Given an open set $\Omega \subset \mathbb{R}^d$, denote $\mathcal{C}^s(\Omega)$ as the class of $s$-times differentiable functions with the norm defined as

$$\|f\|_{\mathcal{C}^s} \triangleq \sum_{i=0}^{s} \left\|f^{(i)}\right\|_\infty. \tag{37}$$

If we want to emphasize that the domain is $\Omega$, we use $\|\cdot\|_{\mathcal{C}^s(\Omega)}$ and $\|\cdot\|_{\infty(\Omega)}$, for the supremum norm. We use $c\Omega$ (with $c > 0$) to denote the set $\{\, c\omega \,:\, \omega \in \Omega \,\}$. In our applications, $\Omega$ would be an interval in $\mathbb{R}$, e.g., $(-b, b)$ for some $b > 0$.

Denote the class of CVF functions that are $s$-smooth function over $\omega$ with bandwidth of $b$ and an $r$-bounded norm by

$$\mathcal{F}_{b,r}^s = \left\{ \tilde{V} \in \mathcal{F}_b \,:\, \|\tilde{V}(\cdot; x)\|_{\mathcal{C}^s((-b,b))} \leq r, \, \forall x \in \mathcal{X} \right\}. \tag{38}$$

This is a subset of $\mathcal{F}_b$ that contains VCFs that are $s$-smooth in the frequency domain with a certain norm of smoothness.

We denote the smoothness norm of $\tilde{V} \in \mathcal{F}_{b,r}^s$ by

$$\left\|\tilde{V}\right\|_{\mathcal{C}^s} = \left\|\tilde{V}\right\|_{\mathcal{C}^s(\Omega)} = \sup_{x \in \mathcal{X}} \left\|\tilde{V}(\cdot; x)\right\|_{\mathcal{C}^s(\Omega)}.$$

We would like to study the function approximation properties of $\mathcal{F}_{b,r}^s$, similar to Theorem 12. For simplicity, we focus only on $\beta$-smooth reward distributions, and not the super smooth ones.

We first provide some intermediate results.

**Proposition 13.** *Suppose that the reward random variable $R(x) \sim \mathcal{R}^\pi(\cdot|x)$ has finite absolute moments up to order $s$, with*

$$\mathbb{E}\left[|R(x)|^i\right] \le m_i^i. \qquad i = 1, \ldots, s$$

*Then, for any $\omega \in \mathbb{R}$, we have*

$$|\tilde{R}^{(i)}(\omega; x)| \le m_i^i. \qquad i = 1, \ldots, s$$

*Proof.* This is the direct consequence of Lemma 8 applied to the random variable $R(x) \sim \mathcal{R}^\pi(\cdot|x)$ i.e., $|\tilde{R}^{(i)}(\omega; x)| = |\mathbb{E}\left[(jR)^i e^{jR\omega}\right]| \le \mathbb{E}\left[|(jR)^i||e^{jR\omega}|\right] = \mathbb{E}\left[|R|^i\right]$. $\qquad\square$

This result guarantees that as long as the reward distribution has $s$ finite absolute moments, the CF $\tilde{R}^\pi$ is $s$-times continuously differentiable too, i.e., $\tilde{R}^\pi(\cdot; x) \in \mathcal{C}^s(\mathbb{R})$. This result along with an argument similar to the proof of Theorem 12 can be used to show that at the first iteration of ACVI, where we would like to approximate $\tilde{R}$ with $\tilde{V}_1$, we can choose $\tilde{V}_1(\cdot; x)$ to be from the smoothness class $\mathcal{C}^s((-b, b))$ and incur a small error. The error depends on the bandwidth $b$. The moment condition shows that $\tilde{R}^\pi \in \mathcal{F}_{b,r}^s$ with $b = \sum_{i=0}^s m_i^i$.

The challenge, however, is to show that after applying the Bellman operator $\tilde{T}^\pi$ to $\tilde{V}_1$ (and to $\tilde{V}_k$ with $k > 1$ in later iterations), it still remains in the same (or similar) smoothness class. If not, our function approximation result would only be applicable for the first iteration of the ACVI.

Let us denote two notations for the supremum norm of $\tilde{V}$. Given a domain $\Omega$ and a fixed $x \in \mathcal{X}$, we use

$$\left\|\tilde{V}(\cdot; x)\right\|_{\infty(\Omega)} = \sup_{\omega \in \Omega} |\tilde{V}(\omega; x)|,$$

if we want to emphasize the domain where the supremum is taken over. We denote the supremum norm of a VCF $\tilde{V}$ over both state and frequency space by

$$\left\|\tilde{V}\right\|_\infty = \left\|\tilde{V}\right\|_{\infty(\Omega),\infty} = \sup_{x \in \mathcal{X}} \sup_{\omega \in \Omega} \left|\tilde{V}(\omega; x)\right|.$$

We use $\|\tilde{V}\|_{\infty(\Omega),\infty}$ whenever we want to emphasize the domain $\Omega$, and use $\|\tilde{V}\|_\infty$ otherwise. This notation should not be confused with $\|\tilde{V}\|_{\infty,p}$ defined in Section 3.1.

To show that applying the Bellman operator $\tilde{T}^\pi$ on a function $\tilde{V}$ in a smoothness class does not take the function outside the class, we study the operator's effect on the smoothness norm, i.e., $\|\tilde{T}^\pi \tilde{V}\|_{\mathcal{C}^s}$. The next intermediate result studies the the effect of taking the $i$-th derivative of $\mathcal{P}^\pi(\cdot|x)\tilde{V}(\gamma\omega; \cdot)$ w.r.t. the frequency parameter $\omega$, and upper bounds it by $\gamma^i\|\tilde{V}^{(i)}\|_{\infty(\gamma\Omega),\infty}$

**Proposition 14.** *Let $\Omega$ be an open interval in $\mathbb{R}$ and $i \in \mathbb{N}_0$ an integer number. Assume that $\tilde{V}(\cdot; x)$ is $i$-times differentiable for all $x \in \mathcal{X}$. For any $x \in \mathcal{X}$, we have*

$$\sup_{\omega \in \Omega} \left|\frac{\mathrm{d}^i}{\mathrm{d}\omega^i} \int \mathcal{P}^\pi(\mathrm{d}y|x)\tilde{V}(\gamma\omega; y)\right| \le \gamma^i \left\|\tilde{V}^{(i)}\right\|_{\infty(\gamma\Omega),\infty}.$$

*Proof.* Note that by the chain rule, we have $\frac{\mathrm{d}\tilde{V}(\gamma\omega;x)}{\mathrm{d}\omega} = \gamma\frac{\mathrm{d}\tilde{V}(u;x)}{\mathrm{d}u}|_{u=\gamma\omega}$. By the repeated application of the chain rule, we get that $\frac{\mathrm{d}^i\tilde{V}(\gamma\omega;x)}{\mathrm{d}\omega^i} = \gamma^i\frac{\mathrm{d}^i\tilde{V}(u;x)}{\mathrm{d}u^i}|_{u=\gamma\omega}$. We have

$$\sup_{\omega\in\Omega}\left|\frac{\mathrm{d}^i}{\mathrm{d}\omega^i}\int\mathcal{P}^\pi(\mathrm{d}y|x)\tilde{V}(\gamma\omega;y)\right| = \sup_{\omega\in\Omega}\left|\int\mathcal{P}^\pi(\mathrm{d}y|x)\frac{\mathrm{d}^i\tilde{V}(\gamma\omega;y)}{\mathrm{d}\omega^i}\right|$$

$$= \sup_{\omega\in\Omega}\left|\gamma^i\int\mathcal{P}^\pi(\mathrm{d}y|x)\frac{\mathrm{d}^i\tilde{V}(u;y)}{\mathrm{d}u^i}\Big|_{u=\gamma\omega}\right|$$

$$\leq \gamma^i\int\mathcal{P}^\pi(\mathrm{d}y|x)\sup_{u\in\gamma\Omega}\left|\frac{\mathrm{d}^i\tilde{V}(u;y)}{\mathrm{d}u^i}\right|$$

$$\leq \gamma^i\left\|\tilde{V}^{(i)}\right\|_{\infty(\gamma\Omega),\infty}.$$

$\square$

The next result shows the effect of applying the Bellman operator on the smoothness norm of a CVF.

**Proposition 15.** *Consider $\Omega$ to be an open interval in $\mathbb{R}$. Suppose that $\tilde{V}(\cdot;x) \in \mathcal{C}^s(\Omega)$ for all $x \in \mathcal{X}$. Assume that there exist finite constants $m_0,\dots,m_s$ such that the absolute moments of the rewards satisfy $\mathbb{E}\left[|R(x)|^i\right] \leq m_i^i$ for all $x \in \mathcal{X}$. Let us denote $m = \max_{i=0,\dots,s} m_i$. We then have*

$$\left\|\tilde{T}^\pi\tilde{V}\right\|_{\mathcal{C}^s(\Omega)} \leq s(m+\gamma)^s\left\|\tilde{V}\right\|_{\mathcal{C}^s(\gamma\Omega)}.$$

*Proof.* Consider the Bellman operator applied to $\tilde{V}$, which is $(\tilde{T}^\pi\tilde{V})(\omega;x) = \tilde{R}(\omega;x)\mathcal{P}^\pi(\cdot|x)\tilde{V}(\gamma\omega;\cdot)$. To simplify the notation, we denote $\tilde{h}(\omega;x) = (\tilde{T}^\pi\tilde{V})(\omega;x)$ and $\tilde{g}(\omega;x) = \mathcal{P}^\pi(\cdot|x)\tilde{V}(\gamma\omega;\cdot)$, so $\tilde{h}(\omega;x) = \tilde{R}(\omega;x)\tilde{g}(\omega;x)$. For any $k = 0,\dots,s$, by the Leibniz product rule we have

$$\tilde{h}^{(k)}(\omega;x) = \sum_{i=0}^k\binom{k}{i}\tilde{R}^{(i)}(\omega;x)\tilde{g}^{(k-i)}(\omega;x).$$

We take the supremum over $\omega\in\Omega$ of both sides, and use Propositions 13 and 14 to get

$$\left\|\tilde{h}^{(k)}(\cdot;x)\right\|_{\infty(\Omega)} \leq \sum_{i=0}^k\binom{k}{i}\left\|\tilde{R}^{(i)}(\cdot;x)\right\|_{\infty(\Omega)}\left\|\tilde{g}^{(k-i)}(\cdot;x)\right\|_{\infty(\Omega)}$$

$$\leq \sum_{i=0}^k\binom{k}{i}\left\|\tilde{R}^{(i)}(\cdot;x)\right\|_{\infty(\mathbb{R})}\left\|\tilde{g}^{(k-i)}(\cdot;x)\right\|_{\infty(\Omega)}$$

$$\leq \sum_{i=0}^k\binom{k}{i}m_i^i\gamma^{k-i}\left\|\tilde{V}^{(k-i)}(\cdot;x)\right\|_{\infty(\gamma\Omega)}.$$

We use $m_i \leq m$ and $\|\tilde{V}^{(i)}(\cdot;x)\|_{\infty(\gamma\Omega)} \leq \|\tilde{V}(\cdot;x)\|_{\mathcal{C}^k(\gamma\Omega)}$ for any $i \leq k$ to get that

$$\left\|\tilde{h}^{(k)}(\cdot;x)\right\|_{\infty(\Omega)} \leq \left\|\tilde{V}(\cdot;x)\right\|_{\mathcal{C}^k(\gamma\Omega)}\sum_{i=0}^k\binom{k}{i}m^i\gamma^{k-i} = \left\|\tilde{V}(\cdot;x)\right\|_{\mathcal{C}^k(\gamma\Omega)}(m+\gamma)^k,$$

where the last equality is because of the binomial theorem. As $m \geq 1$ and $\|\tilde{h}(\cdot;x)\|_{\mathcal{C}^k} \leq \|\tilde{h}(\cdot;x)\|_{\mathcal{C}^s}$ (for any $k \leq s$), we have

$$\left\|\tilde{h}(\cdot;x)\right\|_{\mathcal{C}^s(\Omega)} = \sum_{k=0}^s\left\|\tilde{h}^{(k)}(\cdot;x)\right\|_{\infty(\Omega)} \leq \left\|\tilde{V}(\cdot;x)\right\|_{\mathcal{C}^s(\gamma\Omega)}\sum_{k=0}^s(m+\gamma)^k$$

$$\leq s(m+\gamma)^s\left\|\tilde{V}(\cdot;x)\right\|_{\mathcal{C}^s(\gamma\Omega)}.$$

Taking the supremum over $x \in \mathcal{X}$ from both sides leads to the stated result. $\square$

This result shows that after applying the Bellman operator to a CVF $\tilde{V}$ that has a finite smoothness norm $\|\tilde{V}\|_{\mathcal{C}^s(\Omega)}$, its smoothness norm $\|\tilde{T}^\pi \tilde{V}\|_{\mathcal{C}^s(\Omega)}$ remains finite. The upper bound shows that the smoothness norm might expand by a factor that depends on the absolute moments of the reward distribution and the smoothness degrees $s$.

We can now show a result similar to Theorem 12, but for when we choose the current and the next iteration's VCF from $\mathcal{F}^s_{b,r}$.

**Theorem 16.** *Let $\mathcal{R}$ be a $\beta$-smooth distribution. Assume that the reward distribution has $s$ finite absolute moments satisfying $\max_{i=0,\dots,s} \mathbb{E}\left[|R(x)|^i\right] \le m^i$ for all $x \in \mathcal{X}$. Consider function space $\mathcal{F}^s_{b,r}$ with $b \ge \omega_0$ (cf. Definition 2). We have*

$$\sup_{\tilde{V}' \in \mathcal{F}^s_{b,r}} \ \inf_{\tilde{V} \in \mathcal{F}^s_{b,s(m+\gamma)^s r}} \left\| \tilde{V} - \tilde{T}^\pi \tilde{V}' \right\|_{\infty,p} \le \frac{c_1}{b^{p+\beta}},$$

$$\inf_{\tilde{V} \in \mathcal{F}^s_{b,sm^s}} \left\| \tilde{V} - \tilde{R} \right\|_{\infty,p} \le \frac{c_1}{b^{p+\beta}}.$$

*Proof.* Let $\Omega = (-b, b)$. For any $\tilde{V}' \in \mathcal{F}^s_{b,r}$, consider $(\tilde{T}^\pi \tilde{V}')(\omega; x) = \tilde{R}(\omega; x) \int \mathcal{P}^\pi(\mathrm{d}y|x)\tilde{V}'(\gamma\omega; y)$. Because of the frequency shrinkage $\gamma\omega$ term in $\tilde{V}'(\gamma\omega; y)$, the bandwidth of $\tilde{T}^\pi \tilde{V}'$ is at most $\frac{b}{\gamma}$, i.e., it is zero outside $\gamma^{-1}\Omega$. Proposition 15 shows that

$$\left\| \tilde{T}^\pi \tilde{V}' \right\|_{\mathcal{C}^s(\frac{\Omega}{\gamma})} \le s(m+\gamma)^s \left\| \tilde{V}' \right\|_{\mathcal{C}^s(\Omega)} \le s(m+\gamma)^s r. \tag{39}$$

We truncate the high frequency terms of $\tilde{T}^\pi \tilde{V}'$ in order to have a function $\tilde{V}$ that has a bandwidth of $b$. We define $\tilde{V}(\omega; x) = (\tilde{T}^\pi \tilde{V}')(\omega; x)\mathbb{I}\{\omega \in \Omega\}$. This function has a bandwidth of $b$ by construction. Moreover, its smoothness norm satisfies

$$\left\| \tilde{V} \right\|_{\mathcal{C}^s(\Omega)} = \left\| \tilde{T}^\pi \tilde{V}' \right\|_{\mathcal{C}^s(\Omega)} \le \left\| \tilde{T}^\pi \tilde{V}' \right\|_{\mathcal{C}^s(\frac{\Omega}{\gamma})} \le s(m+\gamma)^s r,$$

where the last inequality is due to (39). Therefore, the function $\tilde{V}$ belongs to $\mathcal{F}^s_{b,r'}$ with $r' = s(m+\gamma)^s r$.

The error function $\tilde{\varepsilon} = \tilde{V} - \tilde{T}^\pi \tilde{V}'$ is $\tilde{\varepsilon}(\omega; x) = (\tilde{T}^\pi \tilde{V}')(\omega; x)\mathbb{I}\{\omega \notin \Omega\}$. As $|\tilde{V}'(\omega; x)| \le 1$ for all $x$ and $\omega$, we have

$$|\tilde{\varepsilon}(\omega; x)| = \left| \mathbb{I}\{\omega \notin \Omega\}\tilde{R}(\omega; x) \int \mathcal{P}^\pi(\mathrm{d}y|x)\tilde{V}^\pi(\gamma\omega; y) \right| \le \mathbb{I}\{\omega \notin \Omega\} \left| \tilde{R}(\omega; x) \right|.$$

The norm of $\tilde{\varepsilon}$ can be upper bounded by

$$\|\tilde{\varepsilon}\|_{\infty,p} = \sup_x \sup_{\omega \in \mathbb{R}-\Omega} \left| \frac{\tilde{R}(\omega; x)}{\omega^p} \right| \le \sup_{|\omega| \ge b} \frac{c_1}{|\omega|^{p+\beta}} = \frac{c_1}{b^{p+\beta}}. \tag{40}$$

As this holds for any $\tilde{V}' \in \mathcal{F}^s_{b,r}$, by taking the supremum over $\mathcal{F}^s_{b,r}$, we obtain the first statement.

The proof of the second statement is similar. We choose $\tilde{V}(\omega; x) = \tilde{R}(\omega; x)\mathbb{I}\{\omega \in \Omega\}$. By construction, its bandwidth is $b$. By the assumption on the absolute moments, Proposition 13 indicates that $\tilde{R}^{(i)}(\omega; x) \le m_i^i \le m^i$ for all $i = 0, \dots, s$, all frequencies $\omega \in \mathbb{R}$, and all states $x \in \mathcal{X}$. Therefore,

$$\left\| \tilde{V} \right\|_{\mathcal{C}^s(\Omega)} = \left\| \tilde{R} \right\|_{\mathcal{C}^s(\Omega)} \le \left\| \tilde{R} \right\|_{\mathcal{C}^s(\mathbb{R})} = \sum_{i=0}^s \left\| \tilde{R}^{(i)} \right\|_\infty \le \sum_{i=0}^s m^i \le sm^s,$$

where we used $m \ge 1$ in the last inequality. This shows that $\tilde{V}$ is in $\mathcal{F}^s_{b,sm^s}$. By computing the norm of $\tilde{\varepsilon} = \tilde{R}(\omega; x)\mathbb{I}\{\omega \notin \Omega\}$, as in (40), we obtain the second statement. $\square$

This theorem shows that for any function $\tilde{V}$ that belongs to $\mathcal{F}^s_{b,r}$, we can find an approximation in a slightly larger function space $\mathcal{F}^s_{b,s(m+\gamma)^s r}$. The approximation error depends on the $\beta$-smoothness of

the reward distribution, the bandwidth $b$ of functions represented by the function space, and the $p$ parameter used in the definition of distance $\| \cdot \|_{\infty,p}$, and it behaves like $O(b^{-(p+\beta)})$. A similar result would hold for super smooth reward distributions, but we omit it here.

This result is comparable to Theorem 12. The main difference is that the approximation space is $\mathcal{F}_{b,r}^s$ (38) instead of $\mathcal{F}_b$ (34), and the target function is limited to $\mathcal{F}_{b,r}^s$ instead of any $\mathcal{V}$. The difference between $\mathcal{F}_{b,r}^s$ and $\mathcal{F}_b$ is in the smoothness regularity of the former function space. The function space $\mathcal{F}_b$ does not impose any smoothness in the frequency domain, and its restriction is only on the bandwidth of the functions. As already mentioned, $\mathcal{F}_b$ is a very large function space and may not allow us to provide a convergence rate for the estimation error. The addition of the smoothness regularity leads to a well-behaving complexity of $\mathcal{F}_{b,r}^s$, which is represented by its covering number. Studying its covering number is the topic of the next section.

# D  Covering number of $\mathcal{F}_{b,r}^s$

We provide a covering number result for the function space $\mathcal{F}_{b,r}^s$, defined in (38). The covering number (and its logarithm, the metric entropy) is a measure of the complexity of a function space, and appears in the analysis of the estimation error [Györfi et al., 2002, van de Geer, 2000]. We restrict our analysis to finite state spaces, i.e, $|\mathcal{X}| < \infty$. Our result, Theorem 19, shows that for a finite state space $\mathcal{X}$, the metric entropy w.r.t. the supremum norm behaves as $\log \mathcal{N}(\varepsilon, \mathcal{F}_{b,r}^s, L_\infty) \leq c|\mathcal{X}|b(\frac{r}{\varepsilon})^{-1/s}$ (and similar for other $L_p$-norms). Interestingly, the covering number w.r.t. $d_{\infty,1}$ behaves as $\log \mathcal{N}(\varepsilon, \mathcal{F}_{b,r}^s, d_{\infty,1}) \leq |\mathcal{X}| \, s \log(\frac{2reb^{\frac{s-1}{2}}}{\varepsilon})$ (and similar w.r.t. $d_{1,1}$), which shows a quite different behaviour, i.e., a logarithmic dependence on $\varepsilon$ as opposed to a polynomial dependence.

To prepare for the main result of this section, we define some notations and state a few auxiliary results. Consider a function $f : \Omega \to \mathcal{C}$ with $\Omega \subset \mathbb{R}$. We denote its extension to $\mathbb{R}$ by $\bar{f}$, i.e.,

$$\bar{f}(\omega) = \begin{cases} f(\omega), & |\omega| < b \\ 0. & |\omega| \geq b \end{cases} \tag{41}$$

Let us consider the set of functions belonging to $\mathcal{C}^s(\Omega)$ with a domain $\Omega$ and $r$-bounded $\mathcal{C}^s(\Omega)$-norm and denote it by $B^s(r; \Omega)$, i.e.,

$$B^s(r; \Omega) \triangleq \left\{ f \in C^s(\Omega) \, : \, \|f\|_{\mathcal{C}^s(\Omega)} \leq r \right\}.$$

We sometimes use $B_b^s(r)$ instead of $B^s(r; (-b, +b))$ with $b > 0$. The value of a CVF at $\omega = 0$ is equal to 1, so in order to discuss functions whose restriction to domain $\Omega$ is smooth and takes the value of 1 at $\omega = 0$, we define

$$\bar{B}^s(r; \Omega) = \left\{ \bar{f} : \mathbb{R} \to \mathbb{C} \, : \, f \in B^s(r; \Omega), f(0) = 1 \right\}.$$

We sometimes use $\bar{B}_b^s(r)$ instead of $\bar{B}^s(r; (-b, +b))$.

The following result, which is based on an already known result on the metric entropy of $\mathcal{C}^s([0, 1])$, provides an upper bound for the metric entropy of $B_b^s(r; (-b, b))$.

**Proposition 17.** *For any $1 \leq p \leq \infty$ and $b \geq 1$, the covering number of $B_b^s(r)$ satisfies*

$$\log \mathcal{N} \left( \varepsilon, B_b^s(r), L_p((-b, b)) \right) \leq cb^{1+\frac{1}{sp}} \left( \frac{r}{\varepsilon} \right)^{\frac{1}{s}},$$

*for a constant $c > 0$ that depends only on $s$ and $p$.*

*Proof.* Consider $f \in B^s(r; \Omega)$ with $\Omega = (-b, b)$. The domain $\Omega$ can be partitioned into $\lceil 2b \rceil$ intervals $\Omega_i$ with length of 1 (in the form of $(-b, b+1], (b+1, b+2], \dots$) such that $\Omega \subset \cup_i \Omega_i$. We denote the restriction of $f$ on each interval by $f_i$, i.e., $f_i(\omega) = f(\omega)\mathbb{I}\{\omega \in \Omega_i\}$. The $\mathcal{C}^s$-norm of $f_i$ over $\Omega_i$ is less than or equal to that of $f$ over $\Omega$, i.e., $\|f_i\|_{\mathcal{C}^s(\Omega_i)} \leq \|f\|_{\mathcal{C}^s(\Omega)}$. Therefore, each $f_i$, after a translation, belongs to $B^s(r; (0, 1]) \subset B^s(r; [0, 1])$.

The covering of $B^s(r; (0, 1])$ for each $i = 1, 2, \dots, 2b$ induces a covering of $B^s(r; \Omega)$. To see this, suppose that $N_\varepsilon$ is an $\varepsilon$-covering set of $B^s(r; (0, 1])$ w.r.t. $L_p((0, 1])$. For any function $f \in B^s(r; \Omega)$,

we can write it as $f(\omega) = \sum_{i=1}^{2b} f_i(\omega)$. For each $i$, pick $f_i' \in N_\varepsilon$ (after a shift of domain so that $\Omega_i$ aligns with $(0,1]$) so that $\int |f_i(\omega) - f_i'(\omega)|^p d\omega \leq \varepsilon^p$ (for $1 \leq p < \infty$) or $\|f_i - f_i'\|_\infty \leq \varepsilon$ (for $p = \infty$). We construct $f'(\omega) = \sum_{i=1}^{2b} f_i'(\omega)$. The $L_p(\Omega)$-norm of the difference between $f'$ and $f$ is $\sqrt[p]{\sum_{i=1}^{2b} \int |f_i'(\omega) - f_i(\omega)|^p d\omega} \leq \sqrt[p]{2b}\varepsilon$ (for $1 \leq p < \infty$) or $\|\sum_{i=1}^{2b}(f_i' - f_i)\|_\infty = \max_{i=1,\ldots,2b} \|f_i' - f_i\|_\infty \leq \varepsilon$ (for $p = \infty$). This shows that we can construct an $\sqrt[p]{2b}\varepsilon$-covering (for $1 \leq p < \infty$) or an $\varepsilon$-covering (for $p = \infty$) of $B^s(r; \Omega)$ based on $\varepsilon$-covering $B^s(r; (0,1])$. The number of choices for each $f_i'$ is $|N_\varepsilon|$, so the number of functions to cover $B_b^s(r)$ is upper bounded by $|N_\varepsilon|^{\lceil 2b \rceil}$. As the covering of $B^s(r; [0,1])$ implies a covering on $B^s(r; (0,1])$, we get that

$$\log \mathcal{N}\left(\varepsilon, B^s(r; \Omega), L_p(\Omega)\right) \leq \lceil 2b \rceil \log \mathcal{N}\left(\varepsilon', B^s(r; [0,1]), L_p([0,1])\right), \tag{42}$$

with $\varepsilon' = \frac{\varepsilon}{\sqrt[p]{2b}}$ for $1 \leq p < \infty$ and $\varepsilon' = \varepsilon$ for $p = \infty$.

Corollary 4.3.38 of Giné and Nickl [2015] shows that

$$\log \mathcal{N}\left(\varepsilon, B^s(r; [0,1]), L_p([0,1])\right) \leq c \left(\frac{r}{\varepsilon}\right)^{1/s},$$

for some constant $c$ that depends only on $s$ and $p$.[8] This along with (42) finish the proof. $\qquad\square$

This result can be generalized to other function spaces. There are two points in the proof where the properties of $\mathcal{C}^s$ are used. The first is that the norm of a function is greater or equal to the norm of that function over a restricted domain. This seems to be true for many other norms. The second is the covering number of $B^s(r; [0,1])$. The same inequality holds for more general function spaces, including Sobolev space $\mathbb{W}^{s,2}([0,1])$ and some Besov spaces with the same order of smoothness $s$, cf. Theorem 4.3.36 and Corollary 4.3.38 of Giné and Nickl [2015].

We can also provide a covering number result for $\bar{B}_b^s(r)$ w.r.t. $d_{\infty,1}$ and $d_{1,1}$, instead of the supremum norm for $B_b^s(r)$ in the previous proposition.

**Proposition 18.** *For any $s \geq 1$ and for any $0 \leq \varepsilon < r$, we have*

$$\log \mathcal{N}\left(\varepsilon, \bar{B}^s(r; [-b, +b]), d_{\infty,1}\right) \leq s \log\left(\frac{2erb^{\frac{s-1}{2}}}{\varepsilon}\right),$$

$$\log \mathcal{N}\left(\varepsilon, \bar{B}^s(r; [-b, +b]), d_{1,1}\right) \leq s \log\left(\frac{4ersb^{\frac{s+1}{2}}}{\varepsilon}\right).$$

*Proof.* Let $\Omega = (-b, +b)$. For any $f \in \bar{B}^s(r; \Omega)$, by the Taylor series expansion around $\omega = 0$, we have that for $\omega \in \Omega$,

$$f(\omega) = 1 + \sum_{i=1}^{s-1} f^{(i)}(0)\frac{w^i}{i!} + f^{(s)}(u)\Big|_{0<u<\omega}\frac{\omega^s}{s!},$$

for some $u \in (0, \omega)$. Here without loss of generality we supposed that $\omega$ is non-negative (otherwise, we could write $\omega < u < 0$). As $f \in \bar{B}^s(r; \Omega)$, its $i$-th derivative $f^{(i)}$ is uniformly bounded by $r$ on $\Omega$ for any $i = 0, \ldots, s$. So we can discretize the interval $[-r, +r]$ and approximate the value of the $f^{(i)}$ terms by the quantized value.

Let us discretize the interval $[-r, +r]$ with resolutions $\varepsilon_1, \ldots, \varepsilon_s$, to be determined, and call the resulting sets $U_1, \ldots U_s$, i.e., $U_i = \{-r, -r + \varepsilon_i, -r + 2\varepsilon_i, \ldots, r - \varepsilon_i\}$. The set $U_i$ has $N_i = |U_i| = \frac{2r}{\varepsilon_i}$ elements. We construct $f'(\omega)$ (for $\omega \in \Omega$) as

$$f'(\omega) = 1 + \sum_{i=1}^{s} a_i \frac{\omega^i}{i!},$$

with $a_i \in U_i$ (for $i = 1, \ldots, s$) being selected so that $|a_i - f^{(i)}(0)| \leq \varepsilon_i$ (for $i = 1, \ldots, s-1$) and $|a_s - f^{(s)}(u)| \leq \varepsilon_s$, which exist by the construction of $U_i$. We use $\bar{f}'$ as the extension of $f'$ from $\Omega$ to $\mathbb{R}$.

Both functions $f'$ and $\bar{f}'$ can be identified by an element of the product set $U_1 \times \cdots \times U_s$, so the number of distinct $f'$ and $\bar{f}'$ constructed this way is the number of elements in the product set $|U_1 \times \cdots \times U_s| = N_1 \times \cdots \times N_s$. We can provide an explicit number on the size of this set as soon as we decide on $\varepsilon_1, \ldots, \varepsilon_s$.

Let us verify that this set provides a covering for $\bar{B}^s(r; \Omega)$ w.r.t. $d_{\infty,1}$ and $d_{1,1}$, with a resolution that depends on $\varepsilon_1, \ldots, \varepsilon_s$. The difference between $f$ and $\bar{f}'$ at any $\omega \in \Omega$ is

$$f(\omega) - \bar{f}'(\omega) = \sum_{i=1}^{s-1} (f^{(i)}(0) - a_i)\frac{w^i}{i!} + \left( f^{(s)}(u)\Big|_{0<u<\omega} - a_s \right)\frac{\omega^s}{s!}.$$

As the value of $f$ and $\bar{f}'$ outside $\Omega$ is zero, we can decompose the $d_{\infty,1}(f, \bar{f}')$ distance as follows:

$$d_{\infty,1}(f, \bar{f}') = \sup_{\omega \in \mathbb{R}} \left| \frac{f(\omega) - \bar{f}'(\omega)}{\omega} \right| = \sup_{\omega \in \Omega} \left| \frac{f(\omega) - \bar{f}'(\omega)}{\omega} \right| + \sup_{\omega \in \mathbb{R} - \Omega} \left| \frac{0 - 0}{\omega} \right|$$

$$= \sup_{\omega \in \Omega} \left| \sum_{i=1}^{s-1} (f^{(i)}(0) - a_i)\frac{w^{i-1}}{i!} + \left( f^{(s)}(u)\Big|_{0<u<\omega} - a_s \right)\frac{\omega^{s-1}}{s!} \right|$$

$$\leq \sum_{i=1}^{s} \varepsilon_i \frac{b^{i-1}}{i!}.$$

Likewise, we can decompose the $d_{1,1}(f, \bar{f}')$ distance as follows:

$$d_{\infty,1}(f, \bar{f}') = \int_{-\infty}^{+\infty} \left| \frac{f(\omega) - \bar{f}'(\omega)}{\omega} \right| \mathrm{d}\omega = \int_{-b}^{+b} \left| \frac{f(\omega) - \bar{f}'(\omega)}{\omega} \right| \mathrm{d}\omega + \int_{\mathbb{R}-\Omega} \left| \frac{0-0}{\omega} \right| \mathrm{d}\omega$$

$$= \int_{-b}^{+b} \left| \sum_{i=1}^{s-1} (f^{(i)}(0) - a_i)\frac{w^{i-1}}{i!} + \left( f^{(s)}(u)\Big|_{0<u<\omega} - a_s \right)\frac{\omega^{s-1}}{s!} \right| \mathrm{d}\omega$$

$$\leq \sum_{i=1}^{s} \frac{\varepsilon_i}{i!} \int_{-b}^{+b} |\omega|^{i-1} \mathrm{d}\omega$$

$$\leq \sum_{i=1}^{s} \varepsilon_i \frac{2b^i}{i \times i!}.$$

By choosing $\varepsilon_i = \frac{\varepsilon}{eb^{i-1}}$ (for the $d_{\infty,1}$ case) and $\varepsilon_i = \frac{\varepsilon}{2eib^i}$ (for the $d_{1,1}$ case), we obtain that

$$d_{\infty,1}(f, \bar{f}'), d_{1,1}(f, \bar{f}') \leq \frac{\varepsilon}{e}\sum_{i=1}^{s}\frac{1}{i!} \leq \varepsilon.$$

This shows that $\bar{f}'$ provides an $\varepsilon$-covering for $\bar{B}^s(r; \Omega)$ w.r.t. $d_{\infty,1}$ and $d_{1,1}$. To count the number of elements in this covering set, note that for the $d_{\infty,1}$ case, we have $N_i = |U_i| = \frac{2r}{\varepsilon_i} = \frac{2erb^{i-1}}{\varepsilon}$, and as a result, the total number of elements of the product set $U_1 \times \cdots \times U_s$ is

$$N_1 \times \cdots \times N_s = \left( \frac{2er}{\varepsilon} \right)^s b^{\frac{s(s-1)}{2}}.$$

Similarly, for the $d_{1,1}$ case, we have $N_i = |U_i| = \frac{2r}{\varepsilon_i} = \frac{4erb^i}{\varepsilon}$, and

$$N_1 \times \cdots \times N_s = s! \left( \frac{4er}{\varepsilon} \right)^s b^{\frac{s(s+1)}{2}}.$$

Taking the logarithm of both sides provides the desired result. $\qquad\square$

To provide a covering number for $\mathcal{F}_{b,r}^s$, we have to specify a distance between functions in $\mathcal{F}_{b,r}^s$. We provide results for two different types of distances. The first is for $L_p$-based norms, and the other is for $d_{\infty,1}$ and $d_{1,1}$ (10).

Given two VCF $\tilde{V}_1, \tilde{V}_2 \in \tilde{V}$, and $p, q \in [1, \infty]$, we define

$$
\left\| \tilde{V}_1 - \tilde{V}_2 \right\|_{L_{q,p}} = \begin{cases} \sqrt[q]{\sum_{x \in \mathcal{X}} \left\| \tilde{V}_1(\cdot; x) - \tilde{V}_2(\cdot; x) \right\|_p^q}, & 1 \le q < \infty \\ \max_{x \in \mathcal{X}} \left\| \tilde{V}_1(\cdot; x) - \tilde{V}_2(\cdot; x) \right\|_p. & q = \infty \end{cases}
$$

With these definition, and equipped with Propositions 17 and 18, we are ready to state and prove our result.

**Theorem 19.** *Consider the function space* (38). *(Part I) For any $1 \le p \le \infty$, $q \in \{p, \infty\}$ and $b \ge 1$, the covering number of $\mathcal{F}_{b,r}^s$ satisfies*

$$
\log \mathcal{N}\left( \varepsilon, \mathcal{F}_{b,r}^s, L_{q,p} \right) \le cb^{1+\frac{1}{sp}} \left( \frac{r}{\varepsilon} \right)^{\frac{1}{s}} \times \begin{cases} |\mathcal{X}|, & q = \infty \\ |\mathcal{X}|^{1+\frac{1}{sp}}. & q = p \end{cases}
$$

*for a constant $c > 0$ that depends only on $s$ and $p$.*

*(Part II) It also holds that for any $0 < \varepsilon < r$,*

$$
\log \mathcal{N}\left( \varepsilon, \mathcal{F}_{b,r}^s, d_{\infty,1} \right) \le |\mathcal{X}| \, s \log \left( \frac{2erb^{\frac{s-1}{2}}}{\varepsilon} \right),
$$

$$
\log \mathcal{N}\left( \varepsilon, \mathcal{F}_{b,r}^s, d_{1,1} \right) \le |\mathcal{X}| \, s \log \left( \frac{4ersb^{\frac{s+1}{2}}}{\varepsilon} \right).
$$

*Proof.* We decompose a function $\tilde{V} \in \mathcal{F}_{b,r}^s$ into $|\mathcal{X}|$ functions, each of which can be constructed based on a member of $B_b^s(r)$ (first part) or $\bar{B}_b^s(r)$ (second part). We then relate the covering number of $\mathcal{F}_{b,r}^s$ to the covering of $B_b^s(r)$ or $\bar{B}_b^s(r)$.

We let $\Omega = (-b, b)$. Recall that given a function $f : \Omega \to \mathcal{C}$, we denote the extension of its domain to $\mathbb{R}$ by $\bar{f}$ (41). We decompose a function $\tilde{V} \in \mathcal{F}_{b,r}^s$ into $|\mathcal{X}|$-functions

$$
\tilde{V}(\omega; x) = \sum_{x_i \in \mathcal{X}} \mathbb{I}\{x = x_i\} \bar{f}_{x_i}(\omega).
$$

For the first part of the result, $\bar{f}_{x_i}$ is an extension of a member of a subset of $B_b^s(r)$ that is 1-bounded and is equal to 1 at $\omega = 0$, i.e., $f_x \in B_b^s(r) \cap \{ f : \omega \to \mathcal{C} : |f(\omega)| \le 1, f(0) = 0 \}$ for all $x \in \mathcal{X}$. For the second part, $\bar{f}_{x_i}$ is a member of $\bar{B}_b^s(r)$.

Let us focus on the first part. Consider an $\varepsilon$-covering set $N_\varepsilon$ of $B_b^s(r)$ w.r.t. $L_p(\Omega)$, which entails that for any $f \in B_b^s(r)$, we can find $f' \in N_\varepsilon$ such that $\|f - f'\|_{L_p(\Omega)} \le \varepsilon$. Consider the product set $N_\varepsilon^\times = \prod_{x_i \in \mathcal{X}} N_\varepsilon$ constructed from each of the $|\mathcal{X}|$ covering sets. Any function $\tilde{V}(\omega; x) = \sum_{x_i \in \mathcal{X}} \mathbb{I}\{x = x_i\} \bar{f}_{x_i}(\omega) \in \mathcal{F}_{b,r}^s$ can be approximated by $\tilde{V}'(\omega; x) = \sum_{x_i \in \mathcal{X}} \mathbb{I}\{x = x_i\} \bar{f}'_{x_i}(\omega)$, with $\bar{f}'_{x_i}$ being an extension of $f'_{x_i} \in N_\varepsilon$ and $f'_{x_i}$ itself is selected to satisfy $\left\| f_{x_i} - f'_{x_i} \right\|_p \le \varepsilon$ (which exists by the definition of the covering set). The function $\tilde{V}'$ is close to $\tilde{V}$, in the $\|\cdot\|_{L_{\infty,p}}$-norm, because

$$
\left\| \tilde{V} - \tilde{V}' \right\|_{L_{\infty,p}} = \left\| \sum_{x_i \in \mathcal{X}} \mathbb{I}\{x = x_i\} \left( \bar{f}_{x_i}(\omega) - \bar{f}'_{x_i}(\omega) \right) \right\|_{L_{\infty,p}}
$$
$$
= \max_{x \in \mathcal{X}} \left\| \bar{f}_x - \bar{f}'_x \right\|_{L_p(\mathbb{R})}
$$
$$
= \max_{x \in \mathcal{X}} \| f_x - f'_x \|_{L_p(\Omega)} \le \varepsilon. \tag{43}
$$

This shows that our choice of $\tilde{V}'$ is $\varepsilon$-close to $\tilde{V}$ w.r.t. $L_{\infty,p}$. Likewise for $1 \le p < \infty$, we have

$$\left\| \tilde{V} - \tilde{V}' \right\|_{p,p} = \left\| \sum_{x_i \in \mathcal{X}} \mathbb{I}\{x = x_i\} \left( \bar{f}_{x_i}(\omega) - \bar{f}'_{x_i}(\omega) \right) \right\|_{p,p}$$

$$= \sqrt[p]{\sum_{x \in \mathcal{X}} \left\| \sum_{x_i \in \mathcal{X}} \mathbb{I}\{x = x_i\} \left( \bar{f}_{x_i} - \bar{f}'_{x_i} \right) \right\|_{L_p(\mathbb{R})}^p}$$

$$= \sqrt[p]{\sum_{x \in \mathcal{X}} \| f_x - f'_x \|_{L_p(\Omega)}^p} \le \sqrt[p]{|\mathcal{X}|}\, \varepsilon. \tag{44}$$

This shows that our choice of $\tilde{V}'$ is $\sqrt[p]{|\mathcal{X}|}\varepsilon$-close to $\tilde{V}$ w.r.t. $L_{p,p}$.

The selected functions $(f'_{x_1}, \ldots, f'_{x_{|\mathcal{X}|}})$ belong to the product space $N_\varepsilon^\times$, which has $|N_\varepsilon|^{|\mathcal{X}|}$ members. The upper bounds (43) and (44) show that this provides an $\varepsilon$-covering of $\mathcal{F}_{b,r}^s$ w.r.t. $L_{\infty,p}$ and a $\sqrt[p]{|\mathcal{X}|}\varepsilon$-covering of $\mathcal{F}_{b,r}^s$ w.r.t. $L_{p,p}$.

To complete the proof of this part, we use Proposition 17, which shows that one can find an $\varepsilon$-covering of $B_b^s(r)$ w.r.t. $L_p(\Omega)$ with $\log |N_\varepsilon| \le cb^{1+\frac{1}{sp}} \left(\frac{r}{\varepsilon}\right)^{\frac{1}{s}}$, with an appropriate choice of $\varepsilon$ for each case.

The proof of the second part of the result is similar too. We construct an $\varepsilon$-cover $N_\varepsilon$ of $\bar{B}_b^s(r)$ w.r.t. $d_{\infty,1}$ (or $d_{\infty,1}$), which entails that for any $\bar{f} \in \bar{B}_b^s(r)$, we can find $\bar{f}' \in N_\varepsilon$ such that $d_{\infty,1}(\bar{f}, \bar{f}') \le \varepsilon$ (or $d_{1,1}(\bar{f}, \bar{f}') \le \varepsilon$).

The product set $N_\varepsilon^\times = \prod_{x_i \in \mathcal{X}} N_\varepsilon$ is an $\varepsilon$-covering set for $\mathcal{F}_{b,r}^s$ w.r.t. $d_{\infty,1}$ (or $d_{1,1}$). To see this, we first notice that any function $\tilde{V} \in \mathcal{F}_{b,r}^s$ can be written as $\tilde{V} = \sum_{x_i \in \mathcal{X}} \mathbb{I}\{x = x_i\}\bar{f}_{x_i}(\omega)$ with $\bar{f}_{x_i} \in \bar{B}_b^r(s)$. The function $\tilde{V}$ can be approximated by $\tilde{V}'(\omega; x) = \sum_{x_i \in \mathcal{X}} \mathbb{I}\{x = x_i\}\bar{f}'_{x_i}(\omega)$ with $(\bar{f}'_{x_1}, \ldots, \bar{f}'_{x_{|\mathcal{X}|}}) \in N_\varepsilon^\times$. This is because for each $x_i$, by the definition of the covering set, one can find a $\bar{f}'_{x_i} \in N_\varepsilon$ such that $d_{\infty,1}(\bar{f}_{x_i}, \bar{f}'_{x_i}) \le \varepsilon$ (or $d_{1,1}(\bar{f}_{x_i}, \bar{f}'_{x_i}) \le \varepsilon$), and hence

$$d_{\infty,1}(\tilde{V}, \tilde{V}') = \max_{x \in \mathcal{X}} \sup_{\omega \in \mathbb{R}} \left| \frac{\sum_{x_i \in \mathcal{X}} \mathbb{I}\{x = x_i\} \left( \bar{f}_{x_i}(\omega) - \bar{f}'_{x_i}(\omega) \right)}{\omega} \right|$$

$$= \max_{x \in \mathcal{X}} d_{\infty,1}\left( \bar{f}_x, \bar{f}'_x \right) \le \varepsilon,$$

$$d_{1,1}(\tilde{V}, \tilde{V}') = \max_{x \in \mathcal{X}} \int \left| \frac{\sum_{x_i \in \mathcal{X}} \mathbb{I}\{x = x_i\} \left( \bar{f}_{x_i}(\omega) - \bar{f}'_{x_i}(\omega) \right)}{\omega} \right|$$

$$= \max_{x \in \mathcal{X}} d_{1,1}\left( \bar{f}_x, \bar{f}'_x \right) \le \varepsilon.$$

The result follows by noticing that the number of elements of $N_\varepsilon^\times$ is $|N_\varepsilon|^{|\mathcal{X}|}$ and then evoking Proposition 18 to provide an upper bound on $|N_\varepsilon|$. $\qquad\square$

The behaviour of these two different covering numbers crucially depends on the choice of the distance. For the $L_{q,p}$ distance, the behaviour as a function of $\varepsilon$ is $O(\varepsilon^{-\frac{1}{s}})$. This is the usual behaviour of the $s$-times differentiable functions over a bounded subset of $\mathbb{R}$. This is not surprising as $\mathcal{F}_{b,r}^s$ is a product space of smoothness class over a bounded frequency domain defined over $|\mathcal{X}|$ states.

The covering number behaviour w.r.t. $d_{\infty,1}$ and $d_{1,1}$, however, is a different story. Its behaviour as a function of $\varepsilon$ is not polynomial, but logarithmic $O(\log(\frac{1}{\varepsilon}))$, hence has a much slower increase as $\varepsilon$ decreases. In other words, measured according to these distances, the function space $\mathcal{F}_{b,r}^s$ is not very complex. One intuition is that the distances $d_{\infty,1}(\tilde{V}_1, \tilde{V}_2) = \sup_x \sup_\omega |\frac{\tilde{V}_1(\omega;x) - \tilde{V}_2(\omega;x)}{\omega}|$ and $d_{1,1}(\tilde{V}_1, \tilde{V}_2) = \sup_x \int |\frac{\tilde{V}_1(\omega;x) - \tilde{V}_2(\omega;x)}{\omega}| \mathrm{d}\omega$ between two VCFs $\tilde{V}_1$ and $\tilde{V}_2$ are less sensitive to high frequency differences between them as the difference is dampened by the $\omega$ in the denominator.

This result alone does not necessarily imply a faster convergence rate for the estimation error term of solving the regression problem (18). One has to study the estimation error more closely to see

whether or not the covering number appearing in the analysis is in fact w.r.t. $d_{\infty,1}$ or $d_{1,1}$ (or $L_{q,p}$ or some other distance). It is in fact likely that the choice of the covering number depends on the choice of $w(\omega)$ in (18). This is a topic of future research.

Finally, we remark that the extension of this result to more general state spaces such as a subset of $\mathbb{R}^d$ requires making assumptions not only on the reward distribution $\mathcal{R}^\pi(\cdot|x)$ at each state $x$ (which we have already done by assuming that it is $\beta$-smooth and has certain moment conditions), but also on how the reward distribution changes as a function of state, e.g., it belongs to a certain smoothness class. This is similar to assumptions required in the analysis of conventional RL methods for large state spaces, e.g., smoothness of the value function [Farahmand, 2011, Farahmand et al., 2016]. That type of analysis is possible to extend to our case too, but we postpone it to a future study.

**Acknowledgments**

I would like to thank the anonymous reviewers for their helpful feedback, particularly Reviewer #4. I acknowledge the funding from the Canada CIFAR AI Chairs program.

## Footnotes

[2]Here $\mathcal{M}(\Omega)$ refers to the space of all probability distributions on an appropriately defined $\sigma$-algebra of $\Omega$, e.g., the Borel $\sigma$-algebra on $\mathbb{R}$. We do not deal with the measure theoretic considerations in this work. Refer to Appendix C of Bertsekas [2013] or Chapter 7 of Bertsekas and Shreve [1978]. We occasionally use $\bar{X}$ to denote the probability distribution $\mu$ of the r.v. $X$.

[3]Here $X$ is a generic r.v. and does not refer to the state. The particular r.v. will be clear from the context.

[4]The metric $d_{\infty,p}$ has been studied under the name of Fourier-based metric Carrillo and Toscani [2007], and is called Toscani distance by Villani [2008].

[5]There are several conventions regarding notations and normalization factors.

[6] This space is larger than the space of feasible VCFs because a CF is uniformly continuous, but $\mathcal{V}$ does not have any continuity restriction.

[7]The proof of this result closely follows the proof of Theorem 2.4 of Hunter and Nachtergaele [2001].

[8] The definition of the $\mathcal{C}^s$-norm of a function $f$ by Giné and Nickl [2015] is $\|f\|_\infty + \|f^{(s)}\|$, which is upper bounded by our definition (37) (see Section 4.3.3 of their book). Therefore, the function space $B_b^s(r; \Omega)$ is a subset of the function space defined with their norm, which is $\{f \in C^s(\Omega) : \|f\|_\infty + \|f^{(s)}\| \leq r\}$. So their covering number is an upper bound on the covering number of the function space we are interested in.