[Reviews · NeurIPS 2019]

Reviewer 1



Thanks to the authors for their response! I maintain that a lot of good work was put in, and primarily recommend including some mention/intuition of scenarios where the proposed method would be preferred over an existing approach (like the examples given in the response). If possible, I think deriving an algorithm based on this foundation and empirically evaluating it (even in a toy domain), would greatly strengthen the paper! ----- The paper had minor grammatical errors. It's clear that a lot of work was put in, and many key theoretical results were worked out for their proposed setting. However, I think the paper strongly lacks in motivation. They suggest that instead of representing a probability distribution over returns, one can instead represent the Fourier transform of it. Can the authors provide any intuition as to why this might be a preferred approach? Are there any known drawbacks of existing distributional RL approaches that this approach can better handle? They show that the characteristic value function in the frequency domain satisfies a Bellman-like equation, which might not be surprising given the linearity of the Fourier transform. Interestingly, it results in a multiplicative form instead of an additive form. Are there any insights as to whether or not this operator might be better behaved compared to the additive form of standard distributional RL? They then characterized the error one might expect from approximately applying value iteration in the frequency domain, and how its implications on errors in the underlying probability distribution. I think this was a good direction as it attempts to tie things back to the standard approach of estimating a probability distribution, but are there any insights on how this might compare to errors in trying to approximate the probability distribution directly? An interesting parallel is that prior work has been done exploring the Fourier transform of returns in the expected, non-distributional setting (De Asis, 2018). While not about the return distribution, perhaps it could give context or motivate why viewing reinforcement learning from the frequency domain might be interesting.

Reviewer 2



Thanks to the author for their response. I think the author should discuss the algorithmic challenges associated with the proposed approach in the main body of the paper. ---------------------------------------------------------------------------------------- Originality - I think the paper is fairly original. I have not seen many applications of Fourier transforms or complex analysis in RL, so the idea of the paper is very novel and fairly exciting. Quality - Overall, I am happy with the quality of the paper. The paper does a good job of laying the theoretical groundwork for using characteristic value function for analysis in RL. For what I could check the Math seemed ok. Clarity - The paper is clear and well-written. Significance - I think the paper can have future significance. I like the paper for its originality. The idea of using characteristic value function is new and unique to me. The paper follows up nicely on the idea of proposing natural extensions like characteristic value functions, value iteration, etc. My main criticism is that I don’t see why it would be advantageous to use CVF instead of just the traditional way of maintaining VF. The paper does not talk about the application of its method. There are no experiments or application of the procedure proposed. This is a major weakness. However, I do believe novel ideas like this are good for conference discussions.

Reviewer 3



POST-REBUTTAL I thank the authors for their detailed response, and in particular the discussion with regards to usefulness of the bounds in the main paper, and possible function approximation classes. I found the discussion around band-limited approximations to the return distribution interesting, and have updated my score from 5 to 6. I'd still argue that further discussion of practical function approximation classes (such as mixtures of Dirac deltas), and/or investigation of a practical algorithm derived from these approximations would make the paper stronger, but am happy to recommend that this paper be accepted, with the additional discussion given by the authors in the rebuttal. --------------------------------------- HIGH-LEVEL COMMENTS This paper proposes a perspective on distributional RL based on characteristic functions. Contractivity of a corresponding Bellman operator is shown under the so-called Toscani metrics, and analysis is undertaken with regards to approximate policy evaluation. The introductory material is well-written, and clearly motivates the authors’ work. Keeping track of return distribution estimates via characteristic functions is an interesting idea, which to my knowledge has not been explored in the distributional RL literature before. My concerns about the significance of the paper stem from the facts that (i) there is no practical algorithm proposed or investigated in the paper. I don’t think this in itself should preclude acceptance, but (ii) there is also little discussion around the usefulness of the theoretical bounds presented in the paper -- it is not clear when the upper bounds presented may be finite, or whether the errors may be driven to zero as the expressivity of the function approximation class appearing in Eqn (14) is increased. Some discussion around why it may be of interest to approximate characteristic functions rather than approximate distributions directly, as in previous DistRL work, would also be interesting. There are some minor mathematical issues regarding conditional independence around Eqn (1) and several later equations, although I don’t believe these affect the overall correctness of the paper significantly. The metrics the authors propose in Section 3.1 allow for distances of +\infty between points. This means that results such as Banach’s fixed point theorem are less readily applicable, and so additional care should be taken around these discussions. As I understand it, it might be necessary to restrict to the case p=1 to recover a unique fixed point using Banach’s fixed point theorem. Eqn (14) seems central to the idea of approximate policy evaluation that the authors consider, but there is little discussion as to how this objective might be optimised in practice (how should the integral over frequencies be approximated?), or with regards to the magnitude of the errors that may arise -- it seems possible that the \infty,p or 1,p norm of the errors might be infinite, which would make the upper bounds appearing in the paper less interesting -- can the authors comment on whether this might be the case, and in what scenarios the error terms might be guaranteed to be finite in norm? I found that the formatting of the supplementary material as an extended version of the main paper made navigating the supplementary material quite difficult. I would ask the authors to consider reformatting the supplementary material to avoid repetition of the main paper where possible in future versions, and to make sure theorem numberings etc. are consistent with the main paper. I have checked the proofs of the results in the main paper. Whilst there are some minor issues, as noted in more detail below, I believe that they are not fundamentally flawed and can be fixed relatively straightforwardly by the authors. DETAILED COMMENTS Line 68 & footnote: if mentioning sigma-algebras in the footnote, they should perhaps be specified in the main paper too. For example, no sigma-algebra is declared for either \mathcal{X} or \mathcal{R}, yet the notation \mathcal{M}(\mathcal{X}) is used. It might be worth mentioning somewhere that you assume the usual Borel sigma algebra for \mathbb{R}, and assume that \mathcal{X} comes equipped with some sigma-algebra. Line 77: it may be slightly more accurate to include a conditioning on X_0 = x on the right-hand side of this equation. Line 82: perhaps remark that G is the sum of two *independent* random variables. Eqn (1): I think the notation here is slightly incorrect. Equality in distribution should apply to random variables, not probability distributions. Similarly, adding two probability distributions does not yield another probability distribution (the distribution on the right-hand side of Eqn (1) would have total mass 1 + \gamma). It would be more correct to say that the random variables concerned are equal in distribution, and that their distributions satisfy a distributional equation (involving e.g. push-forwards and convolutions of measures). This equation is also missing a conditioning on X_0 = x on the right-hand side. Eqn below Line 91: As with Eqn (1), there is an issue with conflating distributions and random variables. Line 109-111 are slightly unclear to me: do the authors mean “probability density function” rather than “probability distribution function”? What about distributions without a pdf? Line 112: why is the notation \mathcal{P}^\pi(\cdot|X=x) used -- isn’t \mathcal{P}^\pi(\cdot|x) more consistent with the definition at the beginning of Line 69? Line 115: I don’t think that R^\pi(x) and G^\p(X’) are conditionally independent given X = x, since they both depend on the action taken in state x. Can the authors comment? It seems that the derivations around Eqn (5) should include a conditioning on the action taken as well; with some care I think this should fix the issue. Line 152 (and later in the paper): I found the notation d_{1/\infty, p} confusing, thinking that 1/\infty was the first parameter of the distance, as opposed to representing two different distances. Lemma 1: in the proof of this result, as with Eqn (5), there should be additional conditioning on the action taken in the initial state to justify conditional independence of the immediate reward and the next state. Line 159: In this paper, the authors allow a metric to take on the value infinity. It may be worth making this clearer in the paper, because this definition means that the notion of contraction is weaker (since \infty <= \gamma \infty), and so results such as Banach’s fixed point theorem do not necessarily apply (for example, if (X,d) is a complete metric space allowing for the possibility of infinite distances between points, then a contraction mapping does not necessarily have a unique fixed point). Value Iteration usually refers specifically to algorithms that aim to find a (near) optimal policy, whereas the algorithm described in Section 4 aims to evaluate a fixed policy. The algorithm might be more accurately described as “characteristic policy evaluation”, or something similar? Line 204: I believe there are similar issues here as with Eqn (5) around lack of conditional independence of R_i and X’_i given X; they are only conditionally independent when conditioning on the action taken at X as well. Line 212: Can the authors comment on whether they would expect the terms (or gradients of the terms) in Eqn (14) to be computable? I wonder whether the integral over frequency is possible to evaluate analytically when \tilde{V} is parametrized according to e.g. a decision tree or a neural network, and if the weighting function w is taken to be uniform, how the integral would be approximated. Line 216: can the authors comment on whether a uniform weighting of frequencies is likely to lead to a finite integral in Eqn (14)? Theorem 2: when are these results likely to be non-vacuous? I.e. when will the \infty, p and/or 1,p norms and metrics be finite? Proof of Theorem 2: in Eqn (24), the final \tilde{\eps} seems to be missing an \omega argument. Line 284: why is this stated as an inequality rather than an equality? Theorem 3: when are these results likely to be non-vacuous? I.e. when will the \infty, p and/or 1,p norms and metrics be finite? Proof of Theorem 3 and related results in the appendix. Proof of Lemma 5: in the first line of Eqn (28), should e^{jwx} be replaced by (e^{jwX_1} - e^{jwX_2})? Line 334: is there a missing factor of 2\pi in the use of Parseval’s theorem here? It seems to appear after Line 337 again. Lines 338-344: there seems to be a problem with a missing factor of 2\pi again. The proof of Lemma 2 (supplementary) would be more neatly expressed as an inductive argument, avoiding the use of ellipses.

[Author Response · NeurIPS 2019]

I am thankful to the reviewers for their careful reading of the paper and their helpful comments. I will fix/revise all minor issues and add an analysis of the function approximation error, which shows that the bounds are non-vacuous. I also emphasize the motivation of this work. Before answering some of the comments in detail, I would like to emphasize that this work opens up a new approach to represent uncertainty of the returns in RL. It provides the fundamental theoretical guarantees that one needs before developing sophisticated algorithms, and empirically evaluating them.

**R4: Non-vacuousness of the bounds in Theorems 2 and 3?**
**A:** The bounds are well-behaving under mild conditions for $p = 1$. It is true that $\|\tilde{\varepsilon}\|_{\infty,p}$ might be infinity for $p > 1$ unless we have restrictive conditions, but its behaviour is reasonable (to be specified) for $p = 1$. Thanks to your comment on this issue, I investigated the approximation error properties for some reasonable choices of $\mathcal{F}$. The result, briefly speaking, is that if the reward distribution is smooth, a band-limited function class $\mathcal{F}_b$ provides an approximation error that goes to zero as $b$ increases. Furthermore, if the first $s$ absolute moments of the reward distribution is finite (uniformly for all $x \in \mathcal{X}$), the CVF $\tilde{V}(\cdot; x)$ belongs to $C^s([-b, b]) \cap \mathcal{F}_b$. This leads to well-behaving covering number, which can be used to obtain a convergence rate for estimation error.

Let us define $\mathcal{F}_b$ as the space of CF with bandwidth of $b$, i.e., $\tilde{V}(\omega; x)$ is zero for $|\omega| > b$. Assume that the reward function is $\beta$-smooth in the sense that $c_0|\omega|^{-\beta} \leq |\tilde{R}(\omega; x)| \leq c_1|\omega|^{-\beta}$ for $|\omega|$ large enough (Jianqing Fan, Annals of Statistics, 1991), which is satisfied by exponential, uniform, gamma, etc. distributions. We can also define super-smooth distributions, with examples such as normal or Cauchy. Let us focus on the approximation error of solving the regression problem in Eq. (14). At each iteration we may pick $\tilde{V}_{k+1}(\omega; x) = (\tilde{T}^\pi \tilde{V}_k)(\omega; x)\mathbb{I}\{\omega \in [-b, +b]\}$. This function is in $\mathcal{F}_b$. Because of the $\beta$-smoothness of $\tilde{R}$, the function approximation error $\tilde{\varepsilon}_{k+1,\text{AE}} = \tilde{V}_{k+1} - \tilde{T}^\pi \tilde{V}_k$ satisfies $\|\tilde{\varepsilon}_{k+1,\text{AE}}\|_{\infty,1} \leq c_1 b^{-(1+\beta)}$ (and faster for super-smooth distributions).

Providing a convergence rate for the estimation error requires some more (mild) assumptions. Let $\mathcal{F}_{b,r}^s$ be the subset of $\mathcal{F}_b$ with the additional condition that $\tilde{V}(\cdot; x) \in C^s([-b, +b])$ (for any fix $x \in \mathcal{X}$). The reasoning required to provide a covering number to be used by the estimation error analysis goes as follows: *(1)* If the reward has $s$-finite absolute moments, its CF $\tilde{R}(\cdot; x)$ is $s$-times differentiable (cf. Lemma 7). *(2)* $\tilde{R}$ can be approximated by a function within $\mathcal{F}_{b,r}^s$, with an error that depends on its $\beta$-smoothness and the choice of $b$ (almost as before). *(3)* We can prove that if $\tilde{V}_k \in \mathcal{F}_{b,r}^s$, it stays in the same smoothness class after applying the Bellman operator (with possibly a larger norm $r'$). *(4)* The estimation error depends on the complexity of $\mathcal{F}_{b,r}^s$. This is a smoothness class, whose covering number is well behaving, i.e., $\log \mathcal{N}(\varepsilon, \mathcal{F}_{b,r}^s) \leq cb(\frac{r}{\varepsilon})^{-1/s}$.

**R1, R3: Motivation? Why not represent the distribution instead?**
**A:** The first motivation is that a new representation opens up possibilities for designing new algorithms. A good example is in the field of control theory, where we have tools to analyze a dynamical system in either the time or frequency domain. Even though they are equivalent in many cases, designing a controller in the frequency domain might be easier. This work brings the frequency-based representation of uncertainty to DistRL. The second motivation is that estimating a probability distribution of returns with a parametric model by performing MLE is infeasible in general (due to the computational challenge of computing the partition function), whereas estimating CF is not (LL39-41).

**R4, R3: How to solve in practice? How Eq. (14) can be solved? How to deal with the integral?**
**A:** Performing ACVI requires us to solve a series of regression problems. Algorithmically the only difference here is that the input includes both state $x$ and frequency $\omega$. Eq. (14) is only one specific (ERM-based) approach, but is not the only one. Focusing on Eq. (14): This is similar to the usual Fitted Q-Iteration. The integral can be approximated numerically, for example by discretizing over various $\omega$. As shown in the response to *R4: Non-vacuousness ...*, we can focus on a bounded domain for $\omega$. I expect computing it analytically might not be possible for general parametrization of CVF, but one might be able to exploit the regularities of, say, a decision tree to compute it more efficiently (constancy of values within a leaf). Also note that estimating ECF has a long history in the statistics and econometrics literature, so it is possible to borrow methods studied there too (see references mentioned in LL36-39).

**R4: Other Q&As. Q:** Conditional independence without action? **A:** The current derivations are correct if the policy is deterministic, as the action is uniquely determined by the state and the policy. If $\pi$ is stochastic, we need to condition on action too, as you mentioned. **Q:** Distribution without density (LL109-111). **A:** CF exists even if the density does not. **Q:** Correct use of Banach fixed point theorem? **A:** When the paper talks about the convergence (LL174-182), I am careful to ensure that we are talking about bounded terms. I will clarify this. **Q:** Missing $\pi$ in Parseval? **A:** You are right! Equality is for a different convention for the Fourier transforms. The change in the final result is that $\pi$ in the denominator becomes $\sqrt{\pi}$. **Q:** L212: Uniform weighting leads to a finite integral? **A:** If we limit the bandwidth to $b$, as discussed earlier, we do not need to be worried about the unboundedness of the integral. More generally, the finiteness seems to depends on the tail behaviour of $\tilde{T}^\pi \tilde{V}_k$, which for example is satisfied with $\beta$-smoothness with large enough $\beta$.

[Meta-Review · NeurIPS 2019]

The reviewers appreciated the authors' response and it addressed most of their concerns. The authors are encouraged to incorporate their additional analysis on the error bounds in the final version, and include intuitions for practical scenarios in which the method could be useful.